# A blueprint for precise and fault-tolerant analog neural networks

Cansu Demirkiran [1] ✉, Lakshmi Nair[2], Darius Bunandar [2] & Ajay Joshi[1]

Analog computing has reemerged as a promising avenue for accelerating deep neural networks (DNNs) to overcome the scalability challenges posed by traditional digital architectures. However, achieving high precision using analog technologies is challenging, as high-precision data converters are costly and impractical. In this work, we address this challenge by using the residue number system (RNS) and composing high-precision operations from multiple low-precision operations, thereby eliminating the need for high-precision data converters and information loss. Our study demonstrates that the RNS-based approach can achieve ≥99% FP32 accuracy with 6-bit integer arithmetic for DNN inference and 7-bit for DNN training. The reduced precision requirements imply that using RNS can achieve several orders of magnitude higher energy efficiency while maintaining the same throughput compared to conventional analog hardware with the same precision. We also present a fault-tolerant dataflow using redundant RNS to protect the computation against noise and errors inherent within analog hardware.

Deep Neural Networks (DNNs) are widely employed across various applications today. Unfortunately, their compute, memory, and communication demands are continuously on the rise. The slow-down in CMOS technology scaling, along with these increasing demands has led analog DNN accelerators to gain significant research interest. Recent research has focused on using various analog technologies such as photonic cores[1–7], resistive arrays[8–12], switched capacitor arrays[13,14], Phase Change Materials (PCM)[15], Spin-Transfer Torque (STT)-RAM[16,17], etc., to enable highly parallel, fast, and efficient matrix-vector multiplications (MVMs) in the analog domain. These MVMs are fundamental blocks used to build general matrix-matrix multiplication (GEMM) operations, which make up more than 90% of the operations in DNN inference and training[18].

The success of this approach, however, is constrained by the limited precision of the digital-to-analog and analog-to-digital data converters (i.e., DACs and ADCs). In an analog accelerator, the data is converted between analog and digital domains using DACs and ADCs before and after every analog operation. Typically, a complete GEMM operation cannot be performed at once in the analog domain due to the fixed size of the analog core. Instead, the GEMM operation is tiled into smaller MVM operations. As a result, each MVM operation produces a partial output that must be accumulated with other partial

outputs to obtain the final GEMM result. Concretely, an MVM operation consists of parallel dot products between $b_w$-bit signed weight vectors and $b_{in}$-bit signed input vectors—each with $h$ elements—resulting in a partial output containing $b_{out}$ bits of information, where $b_{out} = b_{in} + b_w + \log_2(h) - 1$. An ADC with a precision greater than $b_{out}$ (i.e., $b_{ADC} \geq b_{out}$) is required to ensure no loss of information when capturing these partial outputs. Unfortunately, the energy consumption of ADCs increases exponentially with bit precision (often referred to as effective number of bits (ENOB)). This increase is roughly 4 × for each additional bit[19].

As a result, energy-efficient analog accelerator designs typically employ ADCs with lower precision than $b_{out}$ and only capture the $b_{ADC}$ most significant bits (MSBs) from the $b_{out}$ bits of each partial output[20]. Reading only MSBs causes information loss in each partial output leading to accuracy degradation in DNNs, as pointed out by Rekhi et al.[20]. This degradation is most pronounced in large DNNs and large datasets. Figure 1 shows the impact of this approach on DNN accuracy in two tasks: (1) a two-layer convolutional neural network (CNN) for classifying the MNIST dataset[21]: a simple task with only 10 classes, and (2) the ResNet-50 CNN[22] for classifying the ImageNet dataset[23]: a more challenging task with 1000 classes. As the vector size $h$ increases, higher precision is needed at the output to maintain the accuracy in

[1]Boston University, Boston, MA, USA. [2]Lightmatter, Boston, MA, USA. ✉e-mail: cansu@bu.edu

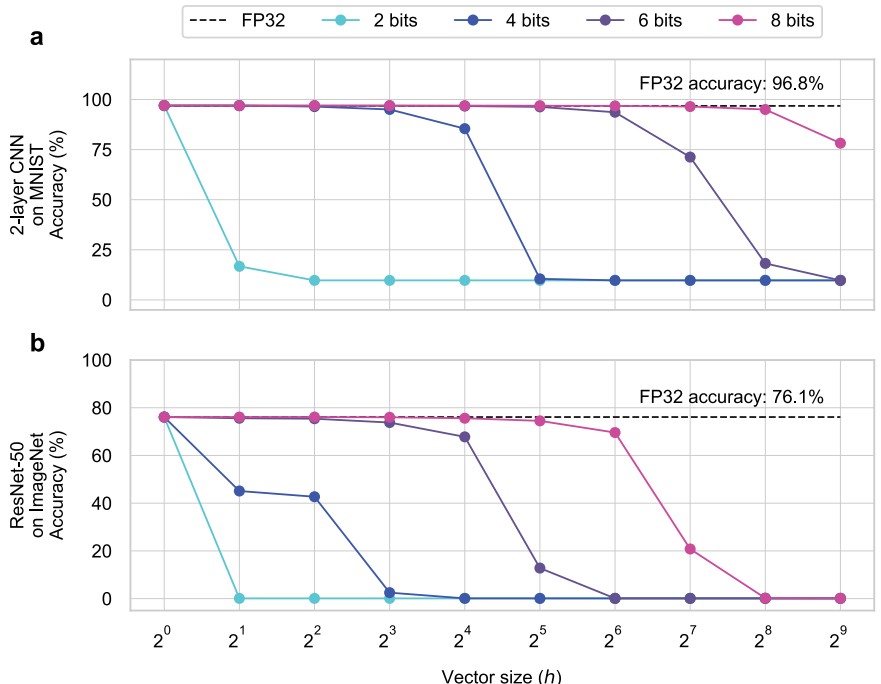

**Fig. 1 | Inference accuracy versus vector size (*h*) for varying data bit-width in a conventional analog core. a** Inference accuracy for a two-layer CNN classifying handwritten digits from the MNIST dataset. **b** Inference accuracy for ResNet-50 classifying images from the ImageNet dataset evaluated in an analog core with varying precision *b* and vector sizes *h*. For both **a** and **b**, *b*-bit precision means $b = b_{DAC} = b_{ADC} < b_{out}$ where *b* varies between 2 and 8. The black dashed line represents the inference accuracy of the same models and datasets in FP32 data format.

both DNNs. Moreover, ResNet-50 experiences accuracy degradation at smaller values of *h* compared to the two-layer CNN. While using a higher precision ADC can help recover from this accuracy degradation, it significantly reduces the energy efficiency of the analog hardware. Essentially, to efficiently execute large DNNs using analog accelerators, it is crucial to find a better way to achieve high accuracy than simply increasing the bit precision of the data converters.

In this work, we present a universal residue number system (RNS)-based framework to overcome the abovementioned challenge in analog DNN inference and training. RNS represents high-precision values using multiple low-precision integer residues for a selected set of moduli. As such, RNS enables high-precision arithmetic without any information loss on the partial products, even when using low-precision DACs and ADCs. The use of RNS leads to a significant reduction in the data converter energy consumption, which is the primary contributor to energy usage in analog accelerators. This reduction can reach up to six orders of magnitude compared to a conventional fixed-point analog core with the same output bit precision.

Our study shows that the RNS-based approach enables ≥99% FP32 inference accuracy by using only 6-bit data converters for state-of-the-art MLPerf (Inference: Datacenters) benchmarks[24] and Large Language Models (LLMs). We also demonstrate the applicability of this approach in training and fine-tuning state-of-the-art DNNs using low-precision analog hardware. The RNS approach, however, is susceptible to noise as small errors in the residues scale up during output reconstruction, leading to larger errors in the standard representation. To address this issue, we incorporate the Redundant RNS (RRNS) error-correcting code[25–27] to introduce fault-tolerance capabilities into the dataflow.

As RNS is closed under multiplication and addition, no significant changes are required in the design of the analog core or in how GEMM operations are performed. Unlike a conventional analog core design, performing RNS operations necessitates an analog modulo operation. This operation can be implemented by using ring oscillators[28] in an analog electrical core or by using optical phase shifters in an analog optical core. Our proposed framework, however, remains agnostic to the underlying technology. Importantly, arbitrary fixed-point precision can be achieved by combining the positional number system (PNS) and RNS in analog hardware. Overall, our presented RNS-based methodology offers a solution combining high accuracy, high energy efficiency, and fault tolerance in analog DNN inference and training.

## Results
### DNN inference and training using RNS
The RNS represents an integer as a set of smaller integer residues. These residues are calculated by performing a modulo operation on the said integer using a selected set of *n* co-prime moduli. Let *A* be an integer. *A* can be represented in the RNS with *n* residues as $\mathcal{A} = \{a_1, \ldots, a_n\}$ for a set of co-prime moduli $\mathcal{M} = \{m_1, \ldots, m_n\}$ where $a_i = |A|_{m_i} \equiv A \bmod m_i$ for $i \in \{1, \ldots, n\}$. *A* can be uniquely reconstructed using the Chinese Remainder Theorem (CRT):

$$A = \left| \sum_{i=1}^{n} a_i M_i T_i \right|_M, \qquad (1)$$

if *A* is within the range [0, *M*) where $M = \prod_i m_i$. Here, $M_i = M/m_i$ and $T_i$ is the multiplicative inverse of $M_i$, i.e., $|M_i T_i|_{m_i} \equiv 1$. Hereinafter, we refer to the integer *A* as the standard representation, while we refer to the set of integers $\mathcal{A} = \{a_1, \ldots, a_n\}$ simply as the residues.

A DNN consists of a sequence of *L* layers. During inference, where the DNN is previously trained and its parameters are fixed, only a forward pass is performed. Generically, the input *X* to ($\ell + 1$)-th layer of a DNN during the forward pass is the output generated by the previous $\ell$-th layer:

$$X^{(\ell+1)} = f^{(\ell)}\left(W^{(\ell)} X^{(\ell)}\right), \qquad (2)$$

where $O^{(\ell)} = W^{(\ell)} X^{(\ell)}$ is a GEMM operation and $f^{(\ell)}(\cdot)$ is an element-wise nonlinear function applied to the GEMM output, $O^{(\ell)}$.

**Table 1 | Data and data converter precision in RNS-based, LP fixed-point, and HP fixed-point analog cores**

| $b_{in}$, $b_w$ | RNS-based Core (This work) | | | | | | LP Fixed-Point Core | | | | HP Fixed-Point Core | | |
|---|---|---|---|---|---|---|---|---|---|---|---|---|---|
| | $b_{DAC}$ | $\log_2 \mathcal{M}$ | $b_{ADC}$ | Moduli Set ($\mathcal{M}$) | RNS Range ($M$) | | $b_{DAC}$ | $b_{out}$ | $b_{ADC}$ | Lost LSBs | $b_{DAC}$ | $b_{out}$ | $b_{ADC}$ |
| 4 | 4 | 4 | 4 | {15, 14, 13, 11} | $\simeq 2^{15}$ | | 4 | 14 | 4 | 10 | 4 | 14 | 14 |
| 5 | 5 | 5 | 5 | {31, 29, 28, 27} | $\simeq 2^{19}$ | | 5 | 16 | 5 | 11 | 5 | 16 | 16 |
| 6 | 6 | 6 | 6 | {63, 62, 61, 59} | $\simeq 2^{24}$ | | 6 | 18 | 6 | 12 | 6 | 18 | 18 |
| 7 | 7 | 7 | 7 | {127, 126, 125} | $\simeq 2^{21}$ | | 7 | 20 | 7 | 13 | 7 | 20 | 20 |
| 8 | 8 | 8 | 8 | {255, 254, 253} | $\simeq 2^{24}$ | | 8 | 22 | 8 | 14 | 8 | 22 | 22 |

DNN training requires both forward and backward passes as well as weight updates. The forward pass in the training is performed the same way as in Eq. (2). After the forward pass, a loss value $\mathcal{L}$ is calculated using the output of the last layer and the ground truth. The gradients of the DNN activations and parameters with respect to $\mathcal{L}$ for each layer are calculated by performing a backward pass after each forward pass:

$$\frac{\partial \mathcal{L}}{\partial X^{(\ell)}} = W^{(\ell)T} \frac{\partial \mathcal{L}}{\partial O^{(\ell)}}, \tag{3}$$

$$\frac{\partial \mathcal{L}}{\partial W^{(\ell)}} = \frac{\partial \mathcal{L}}{\partial O^{(\ell)}} X^{(\ell)T}. \tag{4}$$

Using these weight gradients $\Delta W^{(\ell)} = \frac{\partial \mathcal{L}}{\partial W^{(\ell)}}$, the DNN parameters are updated in each iteration $i$:

$$W_{i+1}^{(\ell)} = W_i^{(\ell)} - \eta \Delta W_i^{(\ell)} \tag{5}$$

with a step size $\eta$ for a simple stochastic gradient descent (SGD) optimization algorithm.

Essentially, for each layer, one GEMM operation is performed in the forward pass and two GEMM operations are performed in the backward pass. Because RNS is closed under addition and multiplication operations, GEMM operations can be performed in the RNS space. Using the RNS, Eq. (2) can be rewritten as:

$$X^{(\ell+1)} = f^{(\ell)} \left( \mathrm{CRT} \left( ||W^{(\ell)}|_{\mathcal{M}} X^{(\ell)}|_{\mathcal{M}}|_{\mathcal{M}} \right) \right). \tag{6}$$

The same approach applies to Eqs. (3) and (4) in the backward pass.

The moduli set $\mathcal{M}$ must be chosen to ensure that the outputs of the RNS operations are smaller than $M$, meaning that

$$\log_2 M \geq b_{out} = b_{in} + b_w + \log_2(h) - 1 \tag{7}$$

should be guaranteed for a dot product between $b_{in}$-bit input and $b_w$-bit weight vectors with $h$-elements. This constraint prevents overflow during computation.

**Precision and accuracy in the RNS-based analog core**
The selection of the moduli set $\mathcal{M}$, which is constrained by Eq. (7), has a direct impact on the achievable precision at the MVM output as well as the energy efficiency of the RNS-based analog core. Table 1 compares RNS-based analog GEMM cores with example moduli sets and regular fixed-point analog GEMM cores with various bit precision. Here, we show two cases for the regular fixed-point representation: (1) the low-precision (LP) case where $b_{out} > b_{ADC} = b_{DAC}$, and (2) the high-precision (HP) case where $b_{out} = b_{ADC} > b_{DAC}$. It should be noted that all three analog cores represent data as fixed-point numbers. We use the term regular fixed-point core to refer to a typical analog core that performs computations in the standard representation (without RNS). RNS-based core refers to an analog core that performs computations on the fixed-point residues.

While the LP approach introduces $b_{out} - b_{ADC}$ bits of information loss in every dot product, the HP approach uses high-precision ADCs to prevent this loss. For the RNS-based core, we picked $b_{in} = b_w = b_{ADC} = b_{DAC} = \lceil \log_2 m_i \rceil \equiv b$ for ease of comparison against the fixed-point cores. Table 1 shows example moduli sets that are chosen to guarantee Eq. (7) for $h = 128$ while keeping the moduli under the chosen bit-width $b$. In this case, for $n$ moduli with bit-width of $b$, $M$ covers $\approx n \cdot b$ bits of range at the output. $h$ is chosen to be 128 as an example considering the common layer sizes in the evaluated MLPerf (Inference: Datacenter) benchmarks. The chosen $h$ provides high throughput with high utilization of the GEMM core.

Figure 2a compares the error (with respect to the FP32 results) observed when performing dot products with the RNS-based core and the LP fixed-point core with the same bit precision. Both cores use the configurations described in Table 1 for the example vector size $h = 128$. The larger absolute error observed in the LP fixed-point case illustrates the impact of the abovementioned information loss due to $b_{ADC} < b_{out}$. HP fixed-point case is not shown as it is equivalent to the RNS case in terms of the observed error.

Figure 2b compares the inference accuracy in MLPerf (Inference: Datacenters) benchmarks[24] and OPT[29] (a transformer-based LLM) when run on an RNS-based analog core and a fixed-point (LP) analog core. The HP fixed-point analog core is not shown as its accuracy is the same as the RNS-based core. The evaluated DNNs, their corresponding tasks, and the datasets are listed in Table 2. Figure 2b shows that the RNS-based approach significantly ameliorates the accuracy drop caused by the low-precision ADCs used in the LP fixed-point approach for all evaluated DNNs. By using the RNS-based approach, it is possible to achieve ≥99% of FP32 accuracy (this cut-off is defined in the MLPerf benchmarks[24]) for all evaluated benchmarks when using residues with 6 bit precision. This number can be lowered to 5 bits for BERT-Large and RNN-T and 4 bits for DLRM.

Besides its success in DNN inference, the RNS-based approach opens the door for analog computing to be used in tasks that require higher precision than DNN inference such as DNN training. Figure 2c–e shows the loss calculated during DNN training/fine-tuning. Table 3 reports the validation accuracy after FP32 and RNS-based low-precision training. Here, the GEMM operations during forward and backward passes of training follow the same methodology as inference, with weight updates carried out in FP32. These experiments show that ≥99% FP32 validation accuracy is achievable after training ResNet-50 from scratch using the RNS-based approach with only 6-bit moduli. Similarly, fine-tuning BERT-Large and OPT-125M by using 5-bit and 7-bit moduli, respectively, can reach ≥99% FP32 validation accuracy. These results are noticeably promising as the previous efforts on analog DNN hardware that adopted the LP fixed-point approach have never successfully demonstrated the training of state-of-the-art DNNs due to the limited precision of this approach.

Figure 3 illustrates the dataflow of the RNS-based analog core when performing MVM as part of the DNN inference/training. An input vector $X$ and a weight matrix $W$ to be multiplied in the MVM unit are first mapped to signed integers. To mitigate the quantization effects, $X$ and each row in $W$ are scaled by an FP32 scaling factor that is unique to

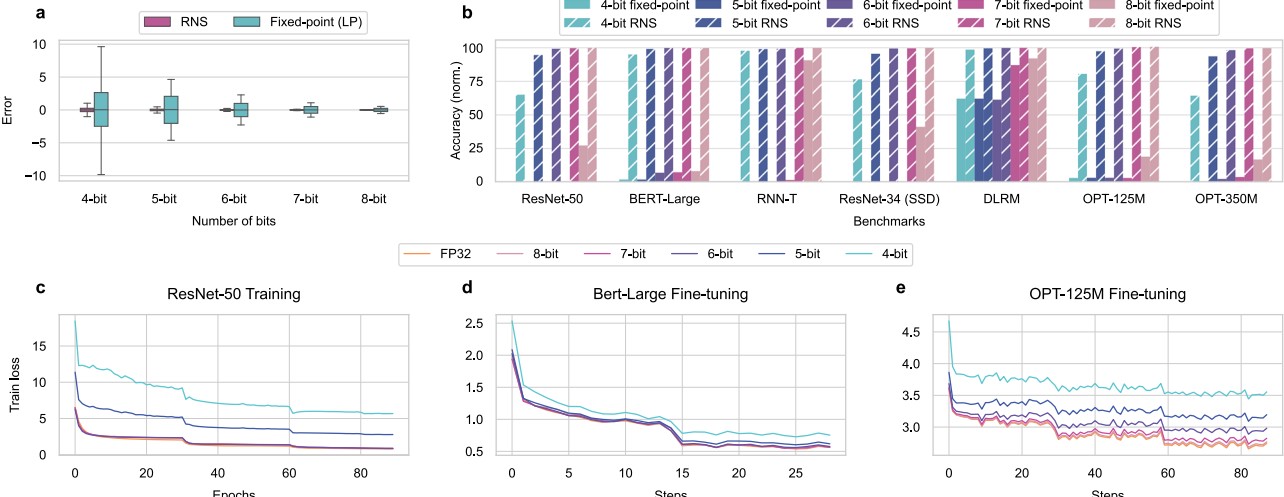

**Fig. 2 | Precision and accuracy comparison of the RNS-based analog core against a regular fixed-point analog core. a** The distribution of average error observed at the output of a dot product performed with the RNS-based analog approach (pink) and the LP regular fixed-point analog approach (cyan). Error is defined as the distance from the result calculated in FP32. The experiments are repeated for 10,000 randomly generated vector pairs with a vector size of $h = 128$. The center lines of the boxes represent the median. The boxes extend between the first and the third quartile of the data, while whiskers extend $1.5 \times$ of the inter-quartile range from the box. **b** Inference accuracy of regular fixed-point (LP) and RNS-based cores (See Table 1) on MLPerf (Inference: Datacenters) benchmarks. The accuracy numbers are normalized by the accuracy achieved in FP32. The bottom three plots show the loss during training for FP32 and the RNS-based approach with varying moduli bit-width. ResNet-50 (**c**) is trained from scratch for 90 epochs using the SGD optimizer with momentum. BERT-Large (**d**) and OPT-125M (**e**) are fine-tuned from pre-trained models. Both models are fine-tuned using the Adam optimizer with a linear learning rate scheduler for 2 and 3 epochs for BERT-Large and OPT-125M, respectively. All inference and training experiments use FP32 for all non-GEMM operations. See Accuracy Modeling under Methods for details.

**Table 2 | MLPerf (Inference: Datacenters) benchmarks**

| DNN | Task | Dataset |
|---|---|---|
| ResNet-50 | Image classification | ImageNet[23] |
| BERT-Large | Question answering | SQuADv1.1[55] |
| RNN-T | Speech recognition | Librispeech[56] |
| ResNet-34 (SSD) | Object detection | MS COCO[57] |
| DLRM | Recommendation | 1TB Click Logs[58] |
| OPT-125M | Language Modeling | Wikitext[59] |
| OPT-350M | Language Modeling | Wikitext |

**Table 3 | Validation accuracy results after training/fine-tuning**

| Precision | ResNet-50 Acc.(%) | BERT-Large F1 Score (%) | OPT-125M Acc.(%)/ Perplexity |
|---|---|---|---|
| FP32 | 75.80 | 91.03 | 43.95/19.72 |
| 8-bit | 75.77 | 90.98 | 43.86/20.00 |
| 7-bit | 75.68 | 90.97 | 43.59/20.71 |
| 6-bit | 75.13 | 90.85 | 42.79/22.62 |
| 5-bit | 59.72 | 90.81 | 41.45/26.17 |
| 4-bit | 42.15 | 89.66 | 38.64/35.65 |

the vector (See Methods). The signed integers are then converted into RNS residues through modulo operation (i.e., forward conversion). By construction, each residue is within the range of $[0, m_i)$. To achieve the same throughput as a fixed-point analog core, the RNS-based analog core with $n$ moduli requires using $n$ analog MVM units—one for each modulus—and running them in parallel. Each analog MVM unit requires a set of DACs for converting the associated input and weight residues into the analog domain. The MVM operations are followed by an analog modulo operation on each output residue vector. Thanks to the modulo operation, the output residues—to be captured by ADCs—are reduced back to the $[0, m_i)$ range. Therefore, a bit precision of $\lceil \log_2 m_i \rceil$ is adequate for both DACs and ADCs to perform input and output conversions without any information loss. The output residues are then converted back to the standard representation in the digital domain using Eq. (1) to generate the signed-integer output vector, which is then mapped back to an FP32 final output $Y$. The non-linear function $f$ (e.g., ReLU, sigmoid, etc.) is then applied digitally in FP32.

**RRNS for fault tolerance**

Analog compute cores are sensitive to noise. In the case of RNS, even small errors in the residues can result in a large error in the corresponding integer they represent. The RRNS[25–27] can detect and correct errors—making the RNS-based analog core fault tolerant. RRNS uses a total of $n + k$ moduli: $n$ non-redundant and $k$ redundant. An RRNS($n + k, n$) code can detect up to $k$ errors and can correct up to $\lfloor \frac{k}{2} \rfloor$ errors. In particular, the error in the codeword (i.e., the $n + k$ residues representing an integer in the RRNS space) can be one of the following cases:

- Case 1: Fewer than $\lfloor \frac{k}{2} \rfloor$ residues have errors—thereby they are correctable,
- Case 2: Between $\lfloor \frac{k}{2} \rfloor$ and $k$ residues have errors or the codeword with more than $k$ errors does not overlap with another codeword in the RRNS space—thereby the error is detectable,
- Case 3: More than $k$ residues have errors and the erroneous codeword overlaps with another codeword in the RRNS space—thereby the error goes undetected.

Errors are detected by using majority logic decoding wherein we divide the total $n + k$ output residues into $\binom{n+k}{n}$ groups with $n$ residues per group and compare the results obtained from each group. If more than 50% of the groups have the same result, then the generated codeword is assumed correct. This either corresponds to Case 1

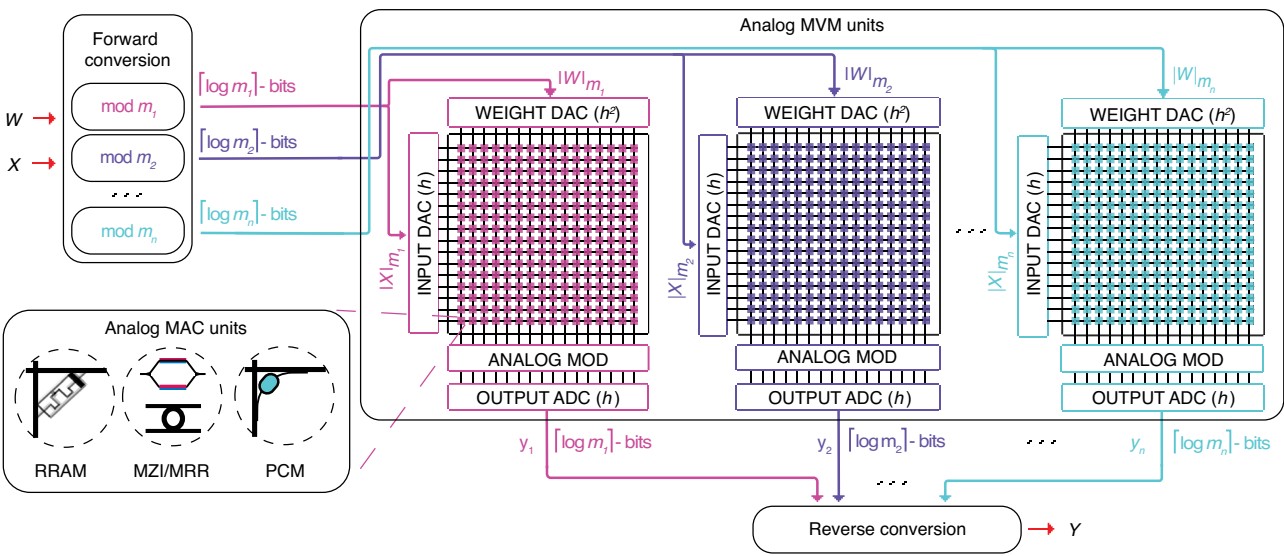

**Fig. 3 | RNS-based analog GEMM dataflow.** The operation is shown for a moduli set $\mathcal{M} = \{m_1, \ldots, m_n\}$. The $n$ $h \times h$ analog MVM units are represented as generic blocks for $n$ moduli. The dataflow is agnostic to the underlying analog technology.

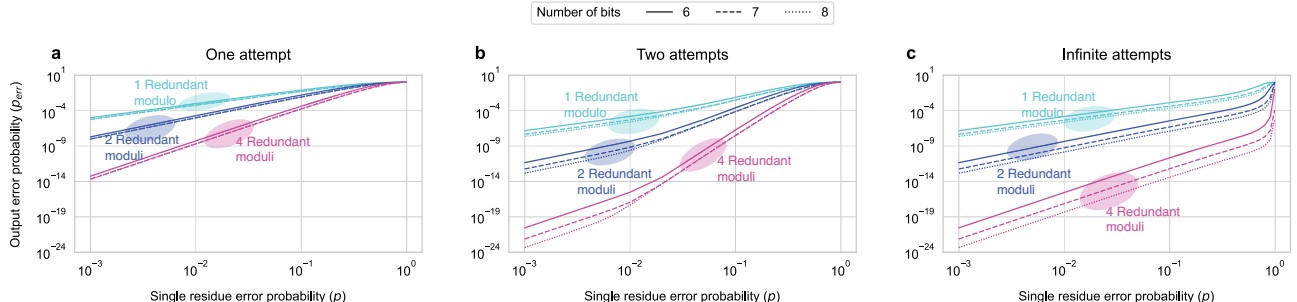

**Fig. 4 | Calculated output error probability ($p_{err}$) versus single residue error probability (p).** $p_{err}$ for one (**a**), two (**b**), and infinite (**c**) error correction attempts and a varying number of redundant moduli ($k$).

where the result is actually correct or Case 3, where the erroneous codeword generated by the majority of the groups overlaps with another codeword. The latter situation leads to an incorrect majority among the groups causing the error to go undetected. In contrast, not having a majority indicates that the generated codeword is erroneous and cannot be corrected. This corresponds to Case 2. In this case, the detected errors can be eliminated by repeating the calculation. One simple way of performing majority logic decoding in this context is to convert the residues in each $\binom{n+k}{n}$ group back to the standard representation via CRT to generate an output value for each group and compare the results. To optimize the hardware performance of this error detection process, more efficient base-extension-based algorithms[30] instead of CRT can be applied.

The final error probability in an RRNS code depends on the non-correctable error probability observed in the residues. The overall error rate is influenced by the chosen moduli set and the number of attempts at correction of the detected errors (See Methods). Let $p_c$, $p_d$, and $p_u$ be the probabilities of Cases 1, 2, and 3 occurring, respectively, when computing a single output. Overall, $p_c + p_d + p_u = 1$. For a single attempt (i.e., $R = 1$), the probability of producing the incorrect output integer is $p_{err}(R = 1) = 1 - p_c = p_u + p_d$. It is possible to repeat the detected erroneous calculations $R > 1$-times to minimize the amount of uncorrected error at the expense of increasing compute latency and energy. In this case, the probability of having an incorrect output after

$R$ attempts of error correction is

$$p_{err}(R) = 1 - p_c \sum_{r=0}^{R-1} (p_d)^r. \tag{8}$$

As the number of attempts increases, the output error probability decreases and converges to $\lim_{R \to \infty} p_{err}(R) = p_u/(p_u + p_c)$.

Figure 4 shows how $p_{err}$ changes with the error probability in a single residue ($p$) for different numbers of redundant moduli ($k$) and attempts ($R$) and moduli sets with different bit-widths. Broadly, as $p$ increases, the $p_{err}$ tends to 1. For a given number of $R$, a higher bit precision and higher $k$ results in a lower $p_{err}$. For a fixed $k$ and a fixed number of bits per moduli, $p_{err}$ decreases as $R$ increases.

Figure 5 investigates the impact of noise on the accuracy of two large and important MLPerf benchmarks—ResNet-50 and BERT-Large—when error correction is applied via RRNS. The two models show similar behavior: increasing $k$ and increasing $R$ decrease $p_{err}$ for the same $p$, enabling to sustain high accuracy for higher $p$. ResNet-50 requires ~3.9 GigaMAC operations (GOp) per inference on a single input image. For a 128 × 128 MVM unit, inferring an ImageNet image through the entire network involves computing ~29.4M partial output elements. Therefore, the expected transition point from an accurate network to an inaccurate network is at $p_{err} = \leq 1/29.4M = 3.4 \times 10^{-8}$. This $p_{err}$ transition point is $\leq 1/358.6M = 2.8 \times 10^{-9}$ for BERT-Large. Figure 5 shows, however, that the evaluated DNNs are more resilient to noise

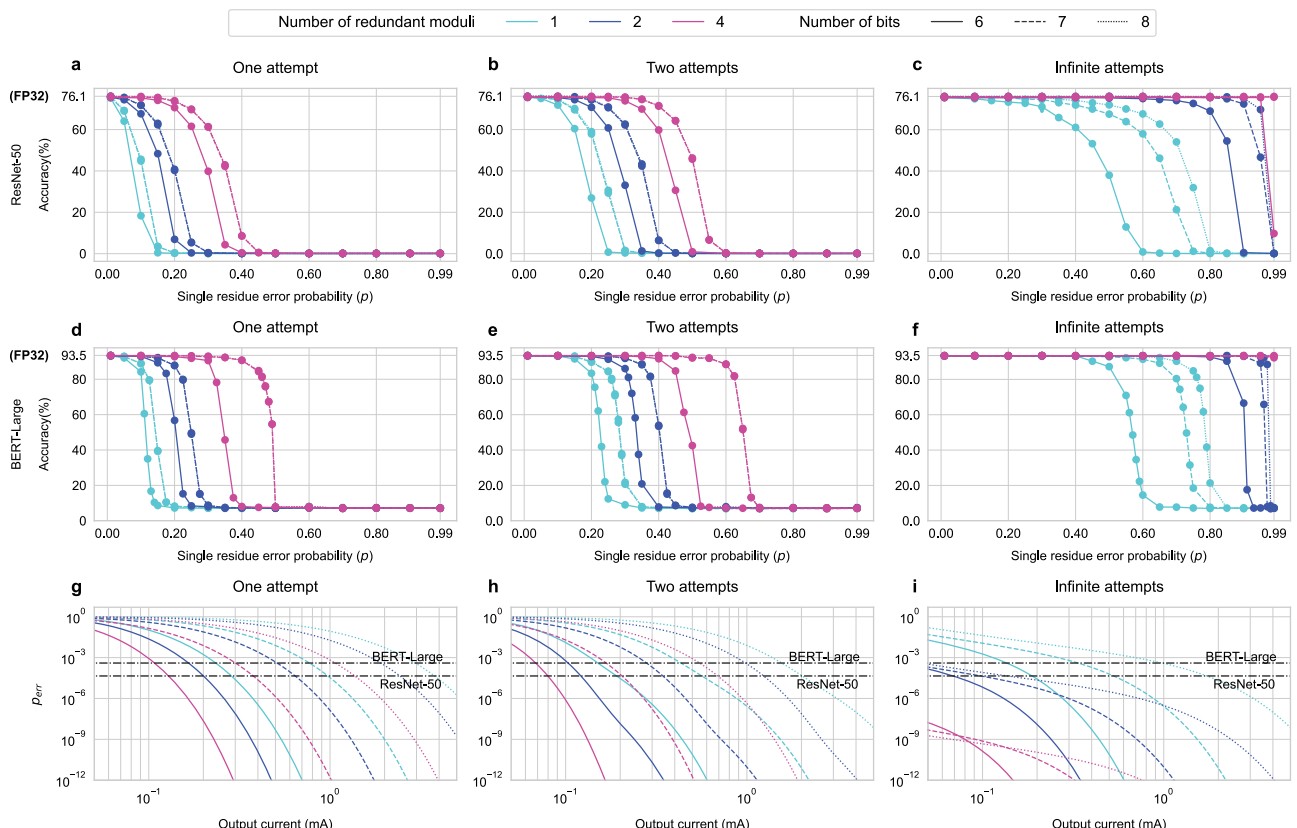

**Fig. 5 | The impact of the single residue error probability (p) on inference accuracy and the required analog current for the fault-tolerant RRNS approach.** The plots show ResNet-50 (**a**–**c**) and BERT-Large (**d**–**f**) inference accuracy under varying $p$ for RRNS with one (**a** and **d**), two (**b** and **e**), and infinite (**c** and **f**) error correction attempts and a varying number of redundant moduli (k). $p_{err}$ caused by shot and thermal noise versus the output current at the photodetector in an analog photonic accelerator for RRNS with one (**g**), two (**h**), and infinite (**i**) error correction attempts and varying $k$. The horizontal black lines show the cut-off points where larger $p_{err}$ starts degrading the accuracy for the evaluated DNNs (i.e., ResNet-50 and BERT-Large).

than expected: they can tolerate higher $p_{err}$ while maintaining good accuracy. The accuracy of ResNet-50 only starts degrading (below 99% FP32) when $p_{err} \approx 4.5 \times 10^{-5}$ (1000 × higher than the estimated value) on average amongst the experiments shown in Fig. 5. This transition probability is $p_{err} \approx 4 \times 10^{-4}$ for BERT-Large (on average 100,000 × higher than the estimated value).

In analog hardware, expected $p$ and $p_{err}$ depend on the underlying analog technology, device characteristics, and many other factors. As an example, we study a photonics-based RNS analog accelerator design that is thermal and shot noise-limited. The noise can be modeled as a Gaussian distribution that is additive to the output value, i.e., $\Sigma_j x_j w_j + \mathcal{N}(0,1)\sigma_{noise}$ for a dot product[31]. Many other noise sources present in various analog designs can be represented using a similar framework. For an analog core where output is captured as an analog current, let us define the maximum achievable current as $I_{out}$, representing the largest output value. A higher $I_{out}$ requires a higher input power, but results in a higher SNR and lower $p_{err}$, creating a tradeoff between power consumption and noise tolerance.

Without any redundant moduli (k = 0), $I_{out} \leq 1$ mA is adequate to prevent accuracy loss due to analog noise in both DNNs (6-bit case). This cut-off is at 2 mA and 8 mA for 7-bit and 8-bit cases, respectively. For instance, for a photonic system, $I_{out} \leq 1$ mA requires ~1 mW (0 dBm) output power (for a photodetector with 1 A/W responsivity)—which is feasible assuming a 10 dBm laser source and 10 dB loss along the optical path.

The required $I_{out}$ can be further lowered by using RRNS. Figure 5g–i shows the relationship between $I_{out}$ and the expected $p_{err}$ for different RRNS. For a smaller number of bits and a higher $k$, a lower $I_{out}$ is adequate to stay under the cut-off $p_{err}$ for the evaluated DNNs.

For example, a 6-bit RRNS with $k = 1$ requires $I_{out} = 0.1$ mA for a single error correction attempt as against the $k = 0$ case where $I_{out} = 1$ mA is needed to avoid accuracy loss due to analog noise. The required $I_{out}$ similarly decreases with the increasing R. Please see Noise Modeling under Methods for details.

## Energy and area efficiency in the RNS-based analog core

Figure 6a shows the energy consumption of DACs and ADCs per dot product for the RNS-based and fixed-point (LP and HP) analog hardware configurations. To achieve the same throughput as the (LP/HP) fixed-point cores, the RNS-based core with $n$ moduli must use $n$ sets of DACs and ADCs. This makes the energy consumption of the RNS-based core $n \times$ larger compared to the LP fixed-point approach. However, the LP fixed-point approach with low-precision ADCs experiences information loss in the partial outputs and hence has lower accuracy.

The RNS-based and HP fixed-point approaches provide the same bit precision (i.e., the same DNN accuracy). Yet, using the RNS-based approach is orders of magnitude more energy-efficient than the HP fixed-point approach. This is mainly because of the high cost of high-precision ADCs required to capture the full output in the HP fixed-point approach. ADCs dominate the energy consumption with approximately three orders of magnitude higher energy usage than DACs with the same bit precision. In addition, energy consumption in ADCs increases exponentially with increasing bit precision[19]. This favors using multiple DACs and ADCs with lower precision in the RNS-based approach over using a single high-precision ADC. The RNS-based approach briefly provides a sweet spot between the LP and HP fixed-point approaches without compromising accuracy and energy efficiency.

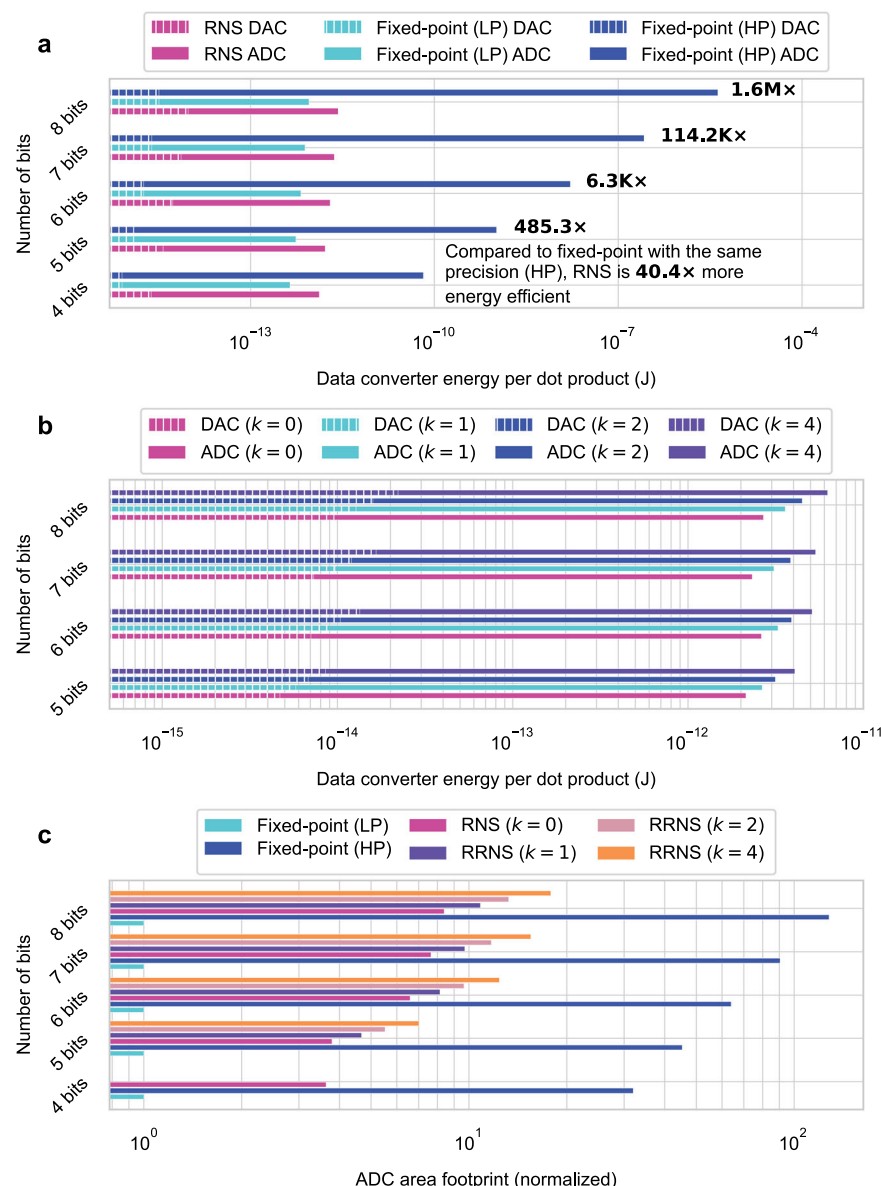

**Fig. 6 | Energy consumption and area comparison of the RNS- and RRNS-based and regular fixed-point analog approaches. a** Energy consumption of data converters (i.e., DACs and ADCs) per dot product for the RNS-based analog approach (pink) and the LP (cyan) and HP (dark blue) regular fixed-point analog approaches. See Data Converter Energy Estimation under Methods for the energy estimation methodology. **b** Energy consumption of data converters per dot product for RNS ($k = 0$) and RRNS ($k = 1, 2,$ and $4$) analog approaches. **c** Normalized area footprint of ADCs for the LP and HP fixed-point, the RNS-based ($k = 0$, $n$ ADCs per dot product), and the RRNS-based ($k > 0$, $n + k$ ADCs per dot product) approaches.

Figure 6b shows the energy consumption of DACs and ADCs when RRNS is used. The plot only shows the 5-to-8-bit cases as there are not enough co-prime moduli smaller than 15 to use RRNS for the 4-bit case. RRNS results in an approximately linear increase in energy consumption as the number of moduli ($n + k$) increases. The compute time does not increase with the increasing $k$ as operations for different moduli are independent and can be performed in parallel.

The area footprint of data converters has a weaker correlation with their bit precision than their energy consumption. In a study by Verhelst and Murmann in 2012[32], the authors observed that the area footprint of ADCs is proportional to $2^{\alpha} b$ where $\alpha \in [0.11, 1.07]$ depending on the type of the ADC, and is $\alpha = 0.5$ when all ADC types are considered. Assuming the same technology node is used, Fig. 6c shows the normalized area footprint of ADCs for the LP and HP fixed-point, RNS ($k = 0$, $n$ ADCs per dot product) and RRNS ($k > 0$, $n + k$ ADCs per dot product) approaches. While the area footprint of the RNS and

RRNS-based approaches are higher than the LP fixed-point approach, they have a smaller area footprint than the HP fixed-point approach for all bit precisions. In addition, the same study points out that the sampling frequency of ADCs is independent of the area footprint. Therefore, in the RNS and RRNS approaches, instead of having multiple ADCs per dot product, one can use a single and faster ADC and perform multiple conversions using the same ADC to achieve the same throughput with better area efficiency.

## Discussion

The RNS (and the fault-tolerant RRNS) framework are agnostic to the analog technology employed. Generally, GEMM operations in the RNS domain can be performed as a regular GEMM operation followed by a modulo operation. Analog GEMM is well-explored in the literature. Previous works leveraged photonics[1-7], crossbar arrays consisting of resistive RAM[8-12], switched capacitors[13,14], PCM cells[15], STT-RAM[16,17], etc.

The analog modulo operation can be performed electrically or optically. As an electrical solution, one can use ring oscillators, a circuit that generates a continuous waveform by cycling through a series of inverters[28], to perform modulo operations. By carefully designing the parameters of the ring oscillator, it is possible to create an output frequency that corresponds to the desired modulus value. Alternatively, the phase of an optical signal can be leveraged for performing modulo due to the periodicity of phases in optical systems. The optical phase is inherently modular against $2\pi$. By modulating the phase of an optical signal, one can achieve modulo operations in the analog domain. See RNS Operations under Methods for details about both approaches. In addition, RNS requires forward and reverse conversion circuits to switch between the RNS and the binary number system (BNS). The forward conversion is a modulo operation while the reverse conversion can be done using the CRT, mixed-radix conversion, or look-up tables. The (digital) hardware costs of these forward and reverse conversion circuits can be reduced by choosing special moduli sets[33,34].

While performing successfully in DNN inference and training, when higher precision is needed, the RNS framework will also be bound by the same precision limitations discussed in this paper for conventional analog hardware. For applications requiring higher-precision arithmetic than the example cases in this study (e.g., some high-performance computing applications, homomorphic encryption, etc.), a higher $M$ value and therefore moduli with higher bit-width might be necessary, requiring higher-precision data converters. This precision limitation can be completely eliminated by combining the RNS framework with PNS, allowing to freely work with arbitrary precision. One can represent an integer value as $D$ separate digits where each digit is represented as a set of residues in the RNS domain and has an RNS range of $M$. This hybrid scheme can achieve $D\log_2 M$ bit precision where $D$ can be liberally increased without increasing the bit precision of the data converters. Different from the RNS-only scheme, the hybrid scheme requires overflow detection and carry propagation from lower digits to higher digits. The overflow detection can be achieved using two sets of residues: primary and secondary. While the operations are performed with both sets of residues, base extension between the two sets helps detect any overflow and propagate the carry to the higher digits if required. See Extended RNS under Methods for details.

RNS is a well-explored numeral system that has been used in a variety of applications including digital signal processing[35], cryptography[36], and DNNs[37,38]. RNS-based DNN computation in digital hardware was proposed for improving energy efficiency by breaking numbers into residues with fewer bits. Res-DNN[38] proposes an RNS-based version of the popular DNN accelerator Eyeriss[39] and RNS-Net[37] uses a processing-in-memory (PIM)-based design and simplifies RNS operations to PIM-friendly ones. A similar work, DNNARA[4], is a nano-photonic (not analog) RNS-based DNN inference accelerator where the authors use $2 \times 2$ optical switches to build a network and manipulate the route of the light through this network to perform multiplication and additions using a one-hot encoded mapping. While all three works are similar to our study in terms of using RNS for DNN inference, we are the first to propose using RNS in the context of analog DNN computation. In addition, these accelerators all propose fully RNS-based dataflows without switching back and forth between RNS and BNS. Although this approach of staying in the RNS domain removes the cost of the RNS-BNS conversions, it requires periodically performing overflow detection and ranging operations in the RNS domain to preserve the integrity of RNS operations. More importantly, these fully RNS-based computations force the end-to-end DNN to be computed in fixed-point arithmetic. Performing nonlinear operations in the RNS domain requires using approximations (e.g., Taylor series expansion) to reduce nonlinear operations into multiply and add operations. These approximations in nonlinear functions cause information loss

and demand higher data precision. As a result, these previous works use 16-bit or higher precision to represent data to achieve high accuracy and their proposals are limited to DNN inference. In our approach, switching back and forth between RNS and BNS for each MVM operation allows us to control the precision of nonlinear operations (which are performed on digital hardware) independently and perform scaling (dynamic quantization) before MVM operations to alleviate the quantization errors at the data converters (See Accuracy Modeling under Methods). This approach also enables us to perform back-propagation and successfully train DNNs with low-precision arithmetic (7-bit) besides DNN inference. In contrast to the few previous analog DNN training demonstrations[40,41] that were limited to very simple tasks (e.g., MNIST classification) and DNNs with a few small layers, our approach can achieve a much higher dynamic range through RNS and can successfully train state-of-the-art DNNs. At last, different from previous works, we analyze the impact of noise on accuracy in RNS-based DNN inference and integrate RRNS to combat the accuracy loss caused by the errors in analog hardware.

In conclusion, our work provides a methodology for precise, energy-efficient, and fault-tolerant analog DNN acceleration. Overall, we believe that RNS is a crucial numeral system for the development of next-generation analog hardware capable of both inference and training of state-of-the-art neural networks for advanced applications, such as generative artificial intelligence.

## Methods

### Handling negative numbers with RNS

An RNS with a dynamic range of $M$ allows representing values within the range of $[0, M)$. This range can be shifted to $[-\psi, \psi]$, where $\psi = \lfloor(M-1)/2\rfloor$, to represent negative values. This is achieved by reassigning the values in between $(0, \psi)$ to be positive, 0 to be zero, and the numbers in between $(\psi, 2\psi)$ to be negative (i.e., $[-\psi, 0)$). Then, the values can be recovered uniquely by using CRT with a slight modification:

$$A = \begin{cases} \sum_{i=1}^{n} |a_i M_i T_i|_M, & \text{if } \sum_{i=1}^{n} |a_i M_i T_i|_M \leq \psi \\ \sum_{i=1}^{n} |a_i M_i T_i|_M - M, & \text{otherwise}. \end{cases} \quad (9)$$

### Data converter energy estimation

The DAC and ADC energy numbers in Fig. 6a, b are estimated by using equations formulated by Murmann[19,42]. The energy consumption of a DAC per $b$-bit conversion is

$$E_{\text{DAC}} = b^2 C_u V_{\text{DD}}^2, \quad (10)$$

where $C_u = 0.5$ fF is a typical unit capacitance and $V_{\text{DD}} = 1$ V is the supply voltage[19]. The energy consumption of an ADC per $b$-bit conversion can be estimated as

$$E_{\text{ADC}} = k_1 b + k_2 4^b. \quad (11)$$

For calculating the coefficients $k_1$ and $k_2$, we used the data from the ADC survey collected by Murmann[42]. The dataset includes all ADC literature published in the two main venues of the field, the International Solid-State Circuits Conference (ISSCC) and the VLSI Circuit Symposium, between the years 1997 and 2023. We removed the data points with a sampling frequency lower than 1 GHz as our design requires high-speed data converters. $k_1$ is calculated as the average of the three samples with the smallest $E_{\text{ADC}}/b$ and $k_2$ as the average of the three samples with the smallest $E_{\text{ADC}}/4^b$ among the available data points[42].

## Accuracy modeling

Both RNS-based and regular fixed-point analog cores are modeled using PyTorch for estimating inference and training accuracy. Convolution, linear, and batched matrix multiplication (BMM) layers are performed as GEMM operations which are computed tile-by-tile as a set of tiled-MVM operations, given the tile size of the analog core. Each input, weight, and output tiles are quantized according to the desired bit precision.

Before quantization, the input vectors and weight tiles are first dynamically scaled at runtime, to mitigate the quantization effects as follows: For an $h \times h$ weight tile $\mathcal{W}_t$, we denote each row vector as $\mathcal{W}_{rt}$ where the subscript $r$ stands for the row and $t$ for the tile. Similarly, an input vector of length $h$ is denoted as $\mathcal{X}_t$ where $t$ indicates the tile. Each weight row $\mathcal{W}_{rt}$ shares a single FP32 scale $s_{rt}^w = \max(|\mathcal{W}_{rt}|)$ and each input vector $\mathcal{X}_t$ shares a single FP32 scale $s_t^x = \max(|\mathcal{X}_t|)$. $h$ scales per $h \times h$ weight tile and one scale per input vector, in total $h + 1$ scales, are stored for each tiled-MVM operation. The tiled MVM is performed between the scaled weight and input vectors, $\widehat{\mathcal{W}}_{rt} = \mathcal{W}_{rt}/s_{rt}^w$ and $\widehat{\mathcal{X}}_t = \mathcal{X}_t/s_t^x$, respectively, to produce $\widehat{Y}_{rt} = \widehat{\mathcal{W}}_{rt}\widehat{\mathcal{X}}_t$. The output $\widehat{Y}_{rt}$ is then quantized (if required) to resemble the output ADCs and multiplied back with the appropriate scales so that the actual output elements $Y_{rt} = \widehat{Y}_{rt} \cdot s_{rt}^w \cdot s_t^x$ are obtained.

Here, the methodology is the same for RNS-based and regular fixed-point cores. For the RNS-based case, in addition to the description above, the quantized input and weight integers are converted into the RNS space before the tiled-MVM operations. MVMs are performed separately for each set of residues and are followed by a modulo operation before the quantization step. The output residues for each tiled MVM are converted back to the standard representation using the CRT.

To accurately model the quantization during forward and backward passes, all GEMM operations (i.e., convolution, linear, and BMM layers) are sandwiched between an input operation $O_{in}$ and an output operation $O_{out}$. This makes the operation order $O_{in}$-GEMM-$O_{out}$ during the forward pass, and $O_{out}$-GEMM-$O_{in}$ in the backward pass. $O_{in}$ quantizes the input and weight tensors in the forward pass and is a null operation in the backward pass. In contrast, $O_{out}$ is a null operation in the forward pass and quantizes the activation gradients in the backward pass. In this way, the quantization is always performed before the GEMM operation. The optimizer (i.e., SGD or Adam) is modified to keep a copy of the FP32 weights to use during the weight updates. Before each forward pass, the FP32 weights are copied and stored. After the forward pass, the quantized model weights are replaced by the previously stored FP32 weights before the step function so that the weight updates are performed in FP32. After the weight update, the model parameters are quantized again for the next forward pass. This high-precision weight update step is crucial for achieving high accuracy in training.

We trained ResNet-50 from scratch by using SGD optimizer for 90 epochs with a momentum of 0.9 and a learning rate starting from 0.1. The learning rate was scaled down by 10 at epochs 30, 60, and 80. We fine-tuned BERT-Large and OPT-125M from the implementations available in the Huggingface transformers repository[43]. We used the Adam optimizer for both models with the default settings. The script uses a linear learning rate scheduler. The learning rate starts at $3e-05$ and $5e-05$ and the models are trained for 2 and 3 epochs, respectively for BERT-Large and OPT-125M.

## Error distribution in the RRNS code space

For an RRNS$(n + k, n)$ with $n$ non-redundant moduli, i.e., $\{(m_1, m_2, \ldots, m_n\}$ and $k$ redundant moduli, i.e., $\{m_{n+1}, m_{n+2}, \ldots, m_{n+k}\}$, the probability distributions, i.e., $p_c$, $p_d$, and $p_u$, of different types of errors, i.e., Case 1, Case 2, and Case 3 that were mentioned in the RRNS for Fault Tolerance subsection are related to the Hamming distance distribution of the RRNS code space. In an RRNS$(n + k, n)$, every integer is represented

as $n + k$ residues ($r_i$ where $i \in \{1, \ldots, n+k\}$) and this vector of $n + k$ residues is considered as an RRNS codeword. A Hamming distance of $\eta \in \{0, 1, \ldots, n+k\}$ between the original codeword and the erroneous codeword indicates that $\eta$ out of $n + k$ residues are erroneous. The erroneous codewords create a new vector space of $n + k$-long vectors where at least one $r_i$ is replaced with $r_i' \neq r_i$ with $i \in \{1, \ldots, n+k\}$ and $r_i' < m_i$. This vector space includes all the RRNS$(n + k, n)$ codewords as well as other possible $n + k$-long vectors that do not overlap with any codeword in the RRNS code space. A vector represents a codeword and is in the RRNS code space if and only if it can be converted into a value within the legitimate range $[0, M]$ of the RRNS$(n + k, n)$ by using the CRT. The number of all vectors that have a Hamming distance $\eta$ from a codeword in RRNS$(n + k, n)$ can be expressed as

$$V_\eta = \sum_{Q\binom{n+k}{\eta}} \prod_{i=1}^{\eta} (m_i - 1), \tag{12}$$

where $Q\binom{n+k}{\eta}$ represents one selection of $\eta$ moduli from $n + k$ moduli while $\sum_{Q\binom{n+k}{\eta}}$ represents the summation over all distinct $\binom{n+k}{\eta}$ selections. The number of codewords that are in the RNS code space with a Hamming distance of $\eta \in \{0, 1, \ldots, n+k\}$ can be expressed as

$$D_\eta = \sum_{h=0}^{\eta-1-k} (-1)^h \binom{n+k-\eta+h}{n+k-\eta} \zeta(n+k, \eta - h), \tag{13}$$

for $k + 1 \le \eta \le n + k$. For $1 \le \eta \le k$, $D_\eta = 0$ and $D_0 = 1$. $\zeta(n+k, \eta)$ represents the total number of non-zero common divisors in the legitimate range $[0, M]$ for any $n + k - \eta$ moduli out of the $n + k$ moduli of the RRNS$(n + k, n)$ code and can be denoted as

$$\zeta(n+k, \eta) = \sum_{Q\binom{n+k}{n+k-\eta}} \left\lfloor \frac{M-1}{m_{i_1} m_{i_2} \ldots m_{i_{(n+k-\eta)}}} \right\rfloor, \tag{14}$$

where $(m_{i_1}, m_{i_2}, \ldots, m_{i_\lambda})$ with $1 \le \lambda \le n + k$ is a subset of the RRNS$(n + k, n)$ moduli set.

An undetectable error occurs only if a codeword with errors overlaps with another codeword in the same RRNS space. Given the distance distributions for the vector space $V$ and the codespace $D$ (Eqs. (12), (13), respectively), the probability of observing an undetectable error ($p_u$) for RRNS$(n + k, n)$ can be computed as

$$p_u = \sum_{\eta=k+1}^{n+k} \frac{D_\eta}{V_\eta} p_E(\eta), \tag{15}$$

where $p_E(\eta)$ is the probability of having $\eta$ erroneous residues in a codeword which can be calculated as

$$p_E(\eta) = \sum_{Q\binom{n+k}{\eta}} p^\eta (1-p)^{(n+k-\eta)}, \tag{16}$$

for a given error probability in a single residue, $p$.

Eq. (13) indicates that for up to $\eta = k$ erroneous residues $D_\eta = 0$, and so an erroneous codeword cannot overlap with another codeword in the RRNS code space. This guarantees the successful detection of

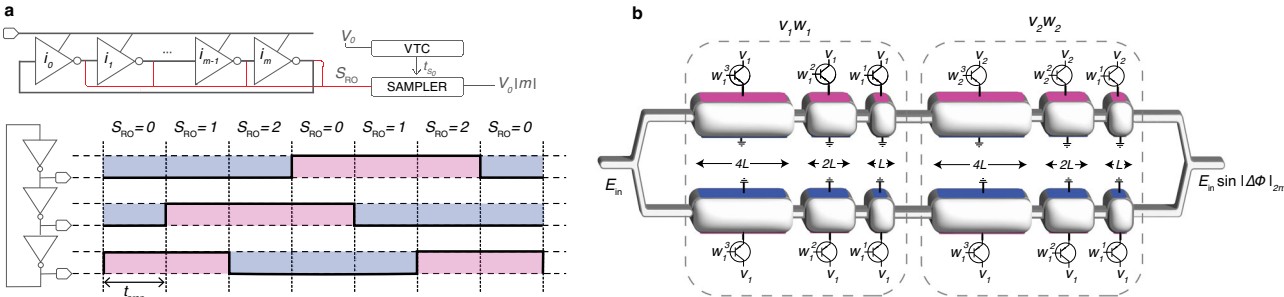

**Fig. 7 | Analog modulo implementations. a** Modulo operation performed using a ring oscillator. A ring oscillator with $N = 3$ inverters is shown to perform modulo against a modulus $m = 3$ as an example. **b** Modular dot product performed using cascaded phase shifters. A modular dot product operation between two 2-element vectors $x$ and $w$, each with 3 digits, is shown in a dual-rail setup. The transistor switch turns on and supplies voltage to the phase shifter when the corresponding digit of $w$ is 1 and it turns off when the corresponding digit of $w$ is 0.

the observed error. If the Hamming distance of the erroneous code-word is $\eta \leq \lfloor \frac{k}{2} \rfloor$, the error can be corrected by the majority logic decoding mechanism. In other words, the probability of observing a correctable error is equal to observing less or equal to $\lfloor \frac{k}{2} \rfloor$ errors in the residues and can be calculated as

$$p_c = \sum_{\eta=0}^{\lfloor \frac{k}{2} \rfloor} p_E(\eta) = \sum_{\eta=0}^{\lfloor \frac{k}{2} \rfloor} \left( \sum_{Q} \binom{n+k}{\eta} p^\eta (1-p)^{(n+k-\eta)} \right). \quad (17)$$

All the errors that do not fall under the undetectable or correctable categories are referred to as detectable but not correctable errors with a probability $p_d$ where $p_d = 1 - (p_c + p_d)$. The equations in this section were collected from the work conducted by Yang[27].

To model the error in the RNS core for the analysis shown in Fig. 5, $p_c$, $p_d$, and $p_u$ are computed for a given RRNS$(n + k, n)$ and $p$ value using Eqs. (15) and (17). Given the number of error correction attempts, $p_{err}$ is calculated according to Eq. (8). Random noise is injected at the output of every tiled-MVM operation using a Bernoulli distribution with a probability of $p_{err}$.

## Noise analysis

In analog hardware, both shot noise and thermal noise can be modeled as Gaussian distributions, i.e., $I_{shot} \sim \sqrt{2q_e \Delta f I_{out}} \mathcal{N}(0,1)$ where $q_e$ is the elementary charge, $\Delta f$ is the bandwidth, $I_{out}$ is the output current of the analog dot product and $I_{thermal} \sim \sqrt{\frac{4k_B \Delta f T}{R_{TIA}}} \mathcal{N}(0,1)$ where $k_B$ is the Boltzmann constant, $T$ is the temperature, and $R_{TIA}$ is the feedback resistor of the transimpedance circuitry.

For a modulus $m$, the consecutive output residues represented in the analog output current should be at least $I_{out}/m$ apart from each other to differentiate $m$ distinct levels. An error occurs in the output residue when $\sqrt{I_{shot}^2 + I_{thermal}^2} \geq I_{out}/2m$ as the residue will be rounded to the next integer otherwise. Therefore, the error probability in a single residue can be calculated as $p = P(\sqrt{2q_e \Delta f I_{out} + \frac{4k_B \Delta f T}{R_{TIA}}} \mathcal{N}(0,1) \geq I_{out}/2m)$. We used $\Delta f = 5$ GHz, $T = 300$ K and $R_{TIA} = 200\Omega$ as typical values in the experiments shown in Fig. 5g–i. For a calculated $p$, $p_{err} = 1 - (1-p)^n$ for an $n$-moduli RNS ($k = 0$). For RRNS ($k > 0$), $p_{err}$ can be obtained using Fig. 4 or Eq. (8).

## RNS operations

The proposed analog RNS-based approach requires modular arithmetic, unlike conventional analog hardware. In this section, we discuss two ways of performing modular arithmetic in the analog domain in

detail. We dive into one electrical solution using ring oscillators and one optical solution using phase shifters.

First, let us consider a ring oscillator with $N$ inverters. In a ring oscillator, where each inverter has a propagation delay of $t_{prop} > 0$, there is always one inverter that has the same input and output—either $1 - 1$ or $0 - 0$—at any given time when the ring oscillator is on. The location of this inverter with the same input and output propagates in the oscillator, along with the signal, every $t_{prop}$ time and rotates due to the ring structure. This rotation forms a modular behavior in the ring when the location of this inverter is tracked.

Let $S_{RO}(t)$ be the state of a ring oscillator where $S_{RO}(t) \in \{0, \ldots, N-1\}$ and $S_{RO}(s) = s$ means that the $s + 1$-th inverter's input and output have the same value at time $t$. $S_{RO}(t)$ keeps rotating between 0 to $N - 1$ as long as the oscillator is on. Fig. 7a shows a simple example where $N = 3$. In the first $t_{prop}$ time interval, the input and output of the first inverter are both 0, therefore, the state $S_{RO}(t < t_{prop}) = 0$. Similarly, when $t_{prop} < t < 2t_{prop}$, the input and output of the second inverter are 1, so $S_{RO}(t_{prop} < t < 2t_{prop}) = 1$. Here, the time between two states following one another (i.e., $t_{prop}$) is fixed and $S_{RO}(t)$ rotates $(0, 1, 2, 0, 1, \ldots)$. Assume the state of the ring oscillator is sampled periodically with a sampling period of $T_s = A \cdot t_{prop}$. Then, the observed change in the state of the ring oscillator between two samples ($S_{RO}(t = T_s) - S_{RO}(t = 0)$) is equivalent to $|A|_N$ where $A$ is a positive integer value. Therefore, to perform modulo with a modulus value $m$, the number of inverters $N$ should be equal to $m$. The dividend number $A$ and the sampling period can be adjusted by changing the analog input voltage to a voltage-to-time converter (VTC).

Here, the dot products can be performed using traditional methods with no change and with any desired analog technology where the output can be represented as an analog electrical signal (e.g., current or voltage) before the analog modulo. The ring oscillator is added to the hardware where the dividend $A$ is the output of the dot product. Here, the total energy consumption of the analog modulo operation depends on $A$ and the area footprint depends on $m$. The ring oscillator typically has a quite smaller energy consumption and area footprint than the other components in the system such as ADCs.

Second, let us consider a typical dual-rail phase shifter. The amount of phase shift introduced by the phase shifter when $v$ and $-v$ voltages are applied on the upper and the bottom arms, respectively, is

$$\Delta\Phi = \frac{vL}{V_{\pi \cdot cm}}, \quad (18)$$

where $V_{\pi \cdot cm}$ is the modulation efficiency of the phase shifter and is a constant value. $\Delta\Phi$ is then proportional to both the length of the shifter $L$ and the amount of applied voltage $v$. Figure 7b shows an example modular dot product operation between two vectors, $x$ and $w$, using cascaded dual-rail phase shifters. This idea is similar to multi-

operand MZIs[44] in which there are multiple active phase shifters controlled by independent signals on each modulation arm. Differently, here, $w$ is encoded digit-by-digit using phase shifters with lengths proportional to $2^j$ where $j$ represents the binary digit number. In the example, each element (i.e., $w_0$ and $w_1$) of the 2-element vector $w$ consists of 3 digits and uses 3 phase shifters, each with lengths $L$, $2L$, and $4L$. If the $j$-th digit of the $i$-th element of $w$, $w_i^j = 1$, a voltage $v_i$ is applied to the phase shifter pair (top and bottom) with the length $2^jL$. If the digit $w_i^j = 0$, then no voltage is applied, and therefore, no phase shift is introduced to the input signal. To encode the second operand $x$, a voltage $v_i$ that is proportional to $x_i$ is applied to all non-zero digits of $w_i$. The multiplication result is then stored in the phase of the propagating signal through the phase shifters, which is modular with $2\pi$. To perform modulo with an arbitrary modulus $m$ instead of $2\pi$, the applied voltage $v$ should be multiplied by the constant $2\pi/m$. For encoding an input integer $x_i$,

$$v_i = x_i \cdot \frac{V_{\pi \cdot cm}}{\pi L} \cdot \frac{2\pi}{m}, \tag{19}$$

should be applied so that the total phase shift at the end of the optical path is

$$\Delta\Phi_{total} = \left|\frac{2\pi}{m}\sum_i\left(\sum_j(2^j w_i^j)x_i\right)\right|_{2\pi} = \frac{2\pi}{m}\left|\sum_i(w_ix_i)\right|_m. \tag{20}$$

The resulting output values in the optical phase are collected at the end of the optical path. These outputs are then re-multiplied by $m/2\pi$ to obtain the outputs of the modular dot products for each modulus.

In the example in Fig. 7b, $w$ is a digital number encoded digit-by-digit to control the phase shifters separately, while $x$ is encoded via an analog voltage $v$. Ideally, the pre-trained $w$ (for inference) can be programmed onto the photonic devices once and kept fixed for multiple inferences. However, today's DNN with millions to billions of parameters makes it impossible to map a whole DNN onto a single accelerator. Therefore, although DNN parameters are not calculated during runtime, $w$ has to be tiled into smaller pieces and loaded into the photonic devices tile by tile. Additionally, modern neural networks that use attention modules require multiplications between matrices that cannot be pre-computed. As a result, both $x$ and $w$ are stored as digital values in the memory before the operations. To this end, the order of these variables can be easily exchanged, i.e., $x$ can be programmed digit-by-digit and $w$ can be used as an analog value or vice versa.

In this approach, the total length of the phase shifter on each arm depends on $m$ and the vector size $h$. Therefore, achieving a feasible design requires a careful selection of the moduli set and the devices used in the design. During an RNS multiplication with modulus $m$ where both $x$ and $w$ are smaller than $m$, the maximum multiplication result is $(m-1)^2$ which can be mapped around zero as $[-\lfloor\frac{(m-1)^2}{2}\rfloor, \lceil\frac{(m-1)^2}{2}\rceil]$. For a modular dot product unit with $h$ elements, the range of the phase shift that the unit can introduce must be within $[-\Delta\Phi_{max}, \Delta\Phi_{max}] = [-\lceil\frac{(m-1)^2}{2}\rceil\frac{2\pi}{m}h, \lceil\frac{(m-1)^2}{2}\rceil\frac{2\pi}{m}h]$, when the maximum bias voltage $v_{max}$ is applied. This requires a total phase shifter length that grows with $O(mh)$ in the dot product unit.

Here, the unit phase shifter length $L$ that creates $\frac{2\pi}{m}$ phase shift is determined by the $V_{\pi \cdot cm}$ of the phase shifter and the maximum bias voltage ($v_{max}$). Essentially, a low $V_{\pi \cdot cm}$ and high $v_{max}$ results in a short device length for the required phase shift. For high-speed phase shifters with modulation bandwidths ≥1 GHz, the most commonly used actuation mechanisms rely on plasma dispersion. For such phase shifters, prior work demonstrated $V_{\pi \cdot cm}$ values lower than 0.5 V·cm[45–49] and optical losses less than 1 dB/cm[50,51].

To determine the total length in this RNS-based approach, the required RNS range (depending on the input precision and vector size)

and the corresponding moduli choice are also critical. A moduli set with fewer but larger values requires fewer but longer dot product units, while a moduli set with more but smaller moduli results in many but shorter dot product units. To quantify, an example moduli set $\{5, 7, 8, 9, 11, 13\}$ can achieve a dynamic range of more than 17 bits-which allows 6-bit arithmetic up to $h = 90$. When a phase shifter with 0.032 V·cm modulation efficiency at 2.8 V is used[48], the phase shifter length varies between 0.3–1.2 mm (per multiplier) for different moduli. With a typical device width of 25 μm, an array size of 64 × 64 (six arrays in total, one 64 × 64 array for each modulus in the abovementioned moduli set) can fit in a typical chip size of 500 mm². This approach is less area efficient and results in higher optical loss per MAC operation compared to a traditional MZI array due to the relatively long phase shifter lengths and utilization of multiple MVM arrays. However, this approach is feasible and it allows us to use lower-precision optical channels (2-to-4-bit for the example above), which can tolerate higher optical loss than even a typical 8-bit photonic hardware while achieving a much higher precision at the output (-17-bit). An equivalent precision requires $2^{17}$ differentiable analog levels at the output of the optical MAC operations and 17-bit ADCs, which is impractical in traditional photonic cores with today's technology (See Fig. 6a).

The scalability of the RNS-based approach can further improve with the developments in photonics technology. Developing high-bandwidth phase shifters with low $V_{\pi \cdot cm}$ and low optical loss is still an active research area. Integration of new materials, e.g., (silicon-)germanium[52], ferroelectrics[50,53], III-V semiconductors[49], 2D materials[54], and organic materials[48], provide promising results despite still being in very early stages. With these integration technologies maturing, more performant silicon photonics phase shifters can enable better area efficiency. In addition, using 3D integration to stack up photonic chiplets (e.g., photonic arrays for different moduli can be implemented on different layers) can further reduce the area footprint in such designs.

## Extended RNS

By combining RNS and PNS, an integer value $Z$ can be represented as $D$ separate digits, $z_d$ where $d \in \{0, 1, \ldots, D-1\}$ and $0 \le z_d < M$:

$$Z = \sum_{d=0}^{D-1} z_d M, \tag{21}$$

and can provide up to $D\log_2 M$ bit precision. This hybrid scheme requires carry propagation from lower digits to higher digits, unlike the RNS-only scheme. For this purpose, one can use two sets of moduli, primary and secondary, where every operation is performed for both sets of residues. After every operation, overflow is detected for each digit and carried over to the next higher-order digit.

Let us define and pick $n_p$ primary moduli $m_i$ where $i \in \{1, \ldots, n_p\}$ and $n_s$ secondary moduli $m_j$ where $j \in \{1, \ldots, n_s\}$, and $m_i \ne m_j \forall \{i, j\}$. Here $M = M_p \cdot M_s = \prod_{i=1}^{n_p} m_i \cdot \prod_{j=1}^{n_s} m_j$ is large enough to represent the largest possible output of the operations performed in this numeral representation and $M_p$ and $M_s$ are co-prime.

In this hybrid number system, operations for each digit are independent of one another and can be parallelized except for the overflow detection and carry propagation. Assume $z_d = z_d|_{p;s}$ consists of primary and secondary residues and is a calculated output digit of an operation before overflow detection. $z_d$ can be decomposed as $z_d|_p = Q_d|_p M_p + R_d|_p$ where $Q_d|_p$ and $R_d|_p$ are the quotient and the remainder of the digit, with respect to the primary RNS. To detect a potential overflow in the digit $z_d$, a base extension from primary to secondary RNS is performed on $z_d|_p$ and the base extended residues are compared with the original secondary residues of the digit, $z_d|_s$. If the residues are the same, this indicates that there is no overflow, i.e., $Q_d|_{p;s} = 0$, and both primary and secondary residues are kept without any carry moved to the next higher digit. In contrast, if the base-

extended secondary residues and the original secondary residues are not the same, there exists an overflow (i.e., $Q_{d|p;s} \neq 0$). In the case of overflow, the remainder of the secondary RNS, $R_{d|s}$, is calculated through a base extension from primary to secondary RNS on $R_{d|p}$ where $R_{d|p} = z_{d|p}$. $Q_{d|s}$ can then be computed as $Q_{d|s} = (z_d|_s - R_d|_s)M_p^{-1}$ where $|M_p \cdot M_p^{-1}|_{M_s} \equiv 1$. $Q_{d|p}$ is calculated through base extension from the secondary to primary RNS on the computed $Q_{d|s}$. The full quotient $Q_{d|p;s}$ is then propagated to the higher-order digit.

Algorithm 1 shows the pseudo-code for handling an operation □ using the extended RNS representation. The operation can be replaced by any operation that is closed under RNS. It should be noted that $z_{d|p;s}$ cannot always be computed as $x_{d|p;s} \square y_{d|p;s}$. For operations such as addition, each digit before carry propagation is computed by simply adding the same digits of the operands, i.e., $z_{d|p;s} = x_{d|p;s} + y_{d|p;s}$. However, for multiplication, each digit of $z_{d|p;s}$ should be constructed as in long multiplication. The multiplication of two numbers in the hybrid number system with $D_x$ and $D_y$ digits requires $D_x D_y$ digit-wise multiplications and the output will result in $D_z = D_x + D_y$ digits in total. Similarly, a dot product is a combination of multiply and add operations. If two vectors with $h$ elements where each element has $D_x$ and $D_y$ digits, the output will require in $D_z = D_x + D_y + \log_2 h$ digits.

**Algorithm 1.** Pseudocode for performing a □ operation using the hybrid number system. Here, $x$ and $y$ are the input operands of □. $z_d$ represents the digits of the output where $d \in \{1, \ldots, D_z\}$, $z_{d|p}$ are the primary residues, and $z_{d|s}$ are the secondary residues. Primary and secondary residues together are referred to as $z_{d'}|_{p;s}$. $Q$ is the quotient and $R$ is the remainder where $z_d = Q_d M_p + R_d$ p2s() and s2p() refer to base extension algorithms from primary to secondary residues and from secondary to primary residues, respectively.

$$Q_{-1}|_{p;s} = 0$$
**for** $d$ in $(0, D_z)$ **do**
$\quad z_{d'}|_{p;s} = (x|_{p;s} \square y|_{p;s})_d$
**end for**
**for** $d$ in $(0, D_z)$ **do**
$\quad z_d|_{p;s} = z_{d'}|_{p;s} + Q_{d-1}|_{p;s}$
$\quad R_d|_p = z_d|_p$
$\quad R_d|_s = \text{p2s}(R_d|_p)$
$\quad$ **if** $R_d|_s = z_{d'}|_s$ **then**
$\quad\quad Q_d|_{p;s} = 0$
$\quad$ **else**
$\quad\quad Q_d|_s = (z_{d'}|_s - R_d|_s)M_p^{-1}$
$\quad\quad Q_d|_p = \text{s2p}(Q_d|_s)$
$\quad$ **end if**
**end for**

### Reporting summary

Further information on research design is available in the Nature Portfolio Reporting Summary linked to this article.

## Data availability

The data that support the plots within this paper and other findings of this study are proprietary to Lightmatter and available under restricted access; access can be obtained from the corresponding authors upon request.

## Code availability

The code that supports the plots within this paper and other findings of this study are proprietary to Lightmatter and available under restricted access; access can be obtained from the corresponding authors upon request.

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

## Acknowledgements
We thank Dr. Rashmi Agrawal and Prof. Vijay Janapa Reddi for their insightful discussions. This work is fully supported by Lightmatter as part of a summer internship.

## Author contributions
D.B. conceived the project idea. C.D. and D.B. developed the theory. C.D. implemented the accuracy modeling and the analytical error models with feedback from D.B. and A.J.; C.D. and L.N. conducted the experiments. D.B. and A.J. supervised the project. C.D. wrote the manuscript with input from all authors.

## Competing interests
The authors declare the following patent application: U.S. Patent Application No.: 17/543,676. L.N. and D.B. declare individual ownership of shares in Lightmatter, a startup company developing photonic hardware for AI. The remaining authors declare no competing interests.
