## [Peer Review File · Nature Communications]

A blueprint for precise and fault-tolerant analog neural networksREVIEWER COMMENTS

Reviewer #1 (Remarks to the Author):

The paper titled "A Blueprint for Precise and Fault-Tolerant Analog Neural Networks" introduces an innovative approach for enhancing the precision and accuracy of DNNs by employing residue number systems (RNS) with CRT. The authors demonstrated that analog accelerators using the RNS-based method can achieve over 99% of FP32 accuracy in state-of-the-art DNN inference, even when employing data converters with 6-bit precision. This study extends the application of the RNS approach to DNN training, where the authors efficiently train DNNs using 7-bit integer arithmetic, all while achieving accuracy comparable to FP32 precision. Additionally, the authors present a fault-tolerant dataflow that incorporates redundant RNS error-correcting codes to safeguard the computation against the noise and errors inherent in an analog accelerator. Finally, the author discusses the implementation of the method in electricity and optics with examples.

In my opinion, whilst the work is executed well, the novelty is incremental and the discussion on hardware implementation is insufficient, therefore, not suitable for a journal like Nature Communication but might benefit from being published in a more specialized journal/conference.

Specific comments:

The structure depicted in Figure 7b (RNS Operations in Optical Domain) appears to resemble a multi-operand MZI. However, there are certain concerns that need to be addressed.

Firstly, when the number of bits in 'w' (referred to as 'j' in the paper) is increased, or when the vector's dimension is expanded, this necessitates a corresponding increase in the total length of the phase shifter. Such an augmentation poses physical design challenges. Therefore, it is crucial for the authors to conduct additional discussion with more data to establish the feasibility of this approach.

Furthermore, the selection of 'w' and 'v' seems somewhat counter-intuitive. Typically, 'w' is pre-trained, while 'x' (or 'v') is often considered variable. However, the authors treat 'w' as a pure digital number, which results in an increased number and physical size of phase shifters. In contrast, 'x' is entirely analog. The paper should delve into a more comprehensive discussion regarding the rationale behind this approach.

Reviewer #2 (Remarks to the Author):

The manuscript entitled "A blueprint for precise and fault-tolerant analog neural networks" by C. Demirkiran et al proposes an RNS approach to process high-precision computations using multiple low-precision operations, in order to reduce the hardware requirements (i.e. high-precision analog-to-digital conversion (ADCs)) and energy cost. This approach is benchmarked in simulation with both DNN inference and training using 6-7 bits precision, achieving a model accuracy comparable to FP32 precision. The authors also analyzed the robustness of the RNS approach against noises and errors using RRNS.

Indeed, the energy consumption and hardware requirement of digital-analog data conversion is a major issue in analog computing. The proposed approach is timely, the motivation of the work is clear and interesting, and the analysis is solid and complete.

However, more clarification is required to differentiate the proposed approach from the prior works. Similar approaches for AI acceleration and digital-analog data conversion are found in "Salamat et al, RNSnet: In-Memory Neural Network Acceleration Using Residue Number System, IEEE International Conference on Rebooting Computing (ICRC) 2018" and "J. Peng, et al, DNNARA: A Deep Neural Network Accelerator using Residue Arithmetic and Integrated Photonics. ICPP: 61:1-61:11 (2020)". Besides, what's new about the RRNS algorithms in this work if they are already developed and used in Ref. 25-27? A detailed comparison to prior works has to be provided.

In addition, some more minor comments:

1. What clock rate is proposed in the calculation of energy consumption?
2. How is k_1 and k_2 calculated in Eq. 11? Is there energy overhead in operating multiple low-speed ADCs?
3. In the redundant RNS, what's the cost (energy or compute time) of using the k redundant moduli to make the system fault tolerant?

Reviewer #3 (Remarks to the Author):

Summary

The authors propose to use RNS for analog MVM to reduce the precision of ADC/DAC while achieving high accuracy for DNN inference/training, so that the energy consumption of analog DNN systems can be reduced.

Pros

- This work attempts to solve the big challenge (impractical ADC) in analog computing and the results are noteworthy. Both inference and training accuracies achieved by the proposed methods with lower-precision ADC are promising.
- There are a good amount of details provided in the methods for the work.

Cons

- The authors only compare the energy consumption and accuracy of different approaches. But it would also be helpful to analyze how much area it takes using RNS compared to conventional HP and LP ADC approach (at iso-throughput setting), I.e., would having n -copy of DAC/ADC take up a significant area of the chip?
- The RNS needs additional modular arithmetic, but the cost of this part is not presented in the paper. It would be interesting to see how much additional energy and area the modular arithmetic takes.
- The authors only analyze the accuracy impact of the proposed error detection/correction scheme using redundant RNS. It would also be interesting to see how much energy and area overhead the redundant RNS takes.
- The $\geq 99\%$ of FP32 accuracy claimed in the paper is achieved without adding noise to the system. It is not discussed in the paper how much accuracy their proposed system can achieve under the typical noise level in an analog circuit.

RESPONSE TO REVIEWERS

We thank the reviewers for their invaluable feedback and insightful suggestions. Below, we first list all the modifications we made to the manuscript and then present our detailed responses to each comment and the specific actions we took to address it thoroughly.

Modifications in the Manuscript

Please note that all the revised or newly added text in the updated manuscript is highlighted in blue for ease of identification.

- We split the “Precision and Energy Efficiency in the RNS-based Analog Core” subsection under the “Results” section into precision and energy efficiency pieces. We merged the precision and accuracy discussions into a single subsection titled “Precision and Accuracy in the RNS-based Analog Core”. We moved the energy efficiency part to the end of the “Results” section with the title “Energy and Area Efficiency in the RNS-based Analog Core”.
- We split Fig. 2 and moved Fig. 2a into Fig. 3 (Fig. 2 in the revised manuscript). We moved Fig. 2b into a new figure (Fig. 6a in the revised manuscript) together with the newly added two plots.
- We added energy overhead analysis for RRNS illustrated in Fig. 6b.
- We added area footprint analysis for RNS and RRNS illustrated in Fig. 6c.
- We included a noise estimation analysis in the Section “Redundant RNS for Fault Tolerance”. The analysis shows the estimated error probability for a analog photonic accelerator. We added Fig. 5(g-i) to illustrate the results.
- We added the energy and area cost of RNS-BNS conversion circuits in the “Discussion” section.
- We added a paragraph comparing our work against related work in RNS-based DNN accelerators to the “Discussion” section.
- We updated the ADC energy estimation formula by adding most recent data to the dataset and using only the high-speed ADC data samples. We added Fig. 7 to illustrate the dataset and the updated fitting function. The updated text and the added figure are in the “Data Converter Energy Estimation” subsection under the “Methods” section. Due to this change in the energy formula, we updated Fig. 2b (Fig. 6a in the revised manuscript).
- We added the “Noise Analysis” subsection under “Methods” to discuss the details of the noise estimation in “Redundant RNS for Fault Tolerance”.
- We modified “Modular Arithmetic with Phase Shifters” subsection under “Methods” and added details on the hardware implementation, required device lengths, and the feasibility of the design.
- We added the energy cost of ring oscillator-based modulo operation to “Modular Arithmetic with Ring Oscillators” subsection under “Methods”.

We hope that we have addressed all comments from reviewers satisfactorily. As such, we hope that the manuscript is now suitable for publication in Nature Communications.

Individual Responses to Reviewer Comments

Reviewer #1:

Reviewer's comment: *"In my opinion, whilst the work is executed well, the novelty is incremental and the discussion on hardware implementation is insufficient, therefore, not suitable for a journal like Nature Communication but might benefit from being published in a more specialized journal/conference."*

Response: We thank the reviewer for their feedback. The two ideas of RNS-based DNN acceleration and analog DNN acceleration have been explored separately in different contexts in previous works [1-5]. However, we believe that our work combines these approaches in a novel way and provides strong contributions to the field. These contributions are listed below.

1. Our work is driven by the limited precision problem in analog hardware due to data converters and achievable signal-to-noise ratio (SNR) in analog circuits. We use RNS to eliminate information loss due to ADCs and achieve high dynamic range during operations in analog compute cores. These ideas have never been presented together and evaluated within a framework for analog DNN computation. RNS has been previously proposed in traditional digital CMOS [1], processing-in-memory (PIM) [2], and nanophotonic [3] DNN accelerators to improve energy efficiency. Similar to our work, these designs break operations into RNS operations with fewer bits. However, we are the first to propose using RNS in the context of analog DNN computation. Our experiments demonstrate that this precision-driven approach helps improve accuracy in DNNs while minimizing the required bit precision of arithmetic operations. We believe that our work is impactful in terms of providing a practical approach to achieving high precision in analog hardware.
2. Although there have been a few attempts to train DNNs using analog technologies [4, 5], these experiments are limited to very small DNNs and simple tasks such as MNIST classification. The limited precision of operations in analog systems prevents such systems from being able to train state-of-the-art DNNs. Our method successfully extends the use cases of analog computing to DNN training and demonstrates the ability to train and finetune real-world DNNs, such as ResNet, BERT, and OPT, using analog hardware.
3. We are the first to evaluate the accuracy of RNS-based DNNs combined with a novel method of mitigating any accuracy loss in the presence of noise using redundant residues. While redundant RNS has been proposed before in the context of error correction code, our paper uses this idea to introduce fault tolerance to analog DNN hardware.

We also added more details on hardware implementation to address the reviewer's concerns about the feasibility of our approach. Please see the details provided in the next question.

Action taken: We added a paragraph about the related work in this field to compare and contrast our work with the related work and highlight our work's novelty. The paragraph is added to the Discussion section on pages 7-8 and copied over below for ease of reference:

“RNS is a well-explored numeral system, which has been used in a variety of applications including digital signal processing, cryptography, and DNNs. RNS-based DNN computation in digital hardware was proposed for improving energy efficiency by breaking numbers into residues with fewer bits. Res-DNN proposes an RNS-based version of the popular DNN accelerator Eyeriss and RNS-Net uses a processing-in-memory (PIM)-based design and simplifies RNS operations to PIM-friendly ones. A similar work, DNNARA, is a nanophotonic (not analog) RNS-based DNN inference accelerator where the authors use 2×2 optical switches to manipulate the route of the light to perform multiplication and additions using a one-hot encoded mapping. While all three works are similar to our study in terms of using RNS for DNN inference, we are the first to propose using RNS in the context of analog DNN computation. In addition, these accelerators all propose fully RNS-based dataflows without switching back and forth between RNS and BNS. Although this approach of staying in the RNS domain removes the cost of RNS-BNS conversions, it requires periodically performing overflow detection and ranging operations in the RNS domain to preserve the integrity of RNS operations. More importantly, these fully RNS-based computations force the end-to-end DNN to be computed in fixed-point arithmetic. Performing nonlinear operations in the RNS domain requires using approximations (e.g., Taylor series expansion) to reduce nonlinear operations into multiply and add operations. These approximations in nonlinear functions cause information loss and demand higher data precision. As a result, these previous works use 16-bit or higher precision to represent data to achieve high accuracy and their proposals are limited to DNN inference. In our approach, switching back and forth between RNS and BNS for each MVM operation allows us to control the precision of nonlinear operations (which are performed on digital hardware) independently and perform scaling (dynamic quantization) before MVM operations to alleviate the quantization errors at the data converters (See Accuracy Modeling under Methods). This approach also enables us to perform backpropagation and successfully train DNNs with low-precision arithmetic besides DNN inference (inference with 6-bit and training with 7-bit). In contrast to the few previous analog DNN training demonstrations that were limited to very simple tasks (e.g., MNIST classification) and DNNs with a few small layers, our approach can achieve a much higher dynamic range through RNS and can successfully train state-of-the-art DNNs. Lastly, different from previous works, we analyze the impact of noise in RNS-based DNN inference and integrate RRNS to combat the accuracy loss caused by the errors in analog hardware.”

Reviewer’s comment: *“The structure depicted in Figure 7b (RNS Operations in Optical Domain) appears to resemble a multi-operand MZI. However, there are certain concerns that need to be addressed.*

Firstly, when the number of bits in 'w' (referred to as 'j' in the paper) is increased, or when the vector's dimension is expanded, this necessitates a corresponding increase in the total length of the phase shifter. Such an augmentation poses physical design challenges. Therefore, it is crucial for the authors to conduct additional discussion with more data to establish the feasibility of this approach.”

Response: It is correct that the optical modular arithmetic approach mentioned in the “Discussion” section is similar to a multi-operand MZI. Different from previous multi-operand MZI works [6], we propose using multiple phase shifters with different lengths (proportional to the

weight of the binary digit, i.e., 2^i) to represent a single number. This enables us to control both operands at runtime by changing the total length of the phase shifter and the applied voltage.

It is also correct that the device length increases with the increasing modulus value m and vector size h . For a single multiplication unit where both x and w are smaller than m , the maximum multiplication result is $(m - 1)^2$ which can be mapped around zero as $[-\lfloor \frac{(m-1)^2}{2} \rfloor, \lfloor \frac{(m-1)^2}{2} \rfloor]$. Therefore, the phase shift range that the unit can introduce must be $[-\Delta\Phi_{max}, \Delta\Phi_{max}] = [-\lfloor \frac{(m-1)^2}{2} \rfloor \frac{2\pi}{m}, \lfloor \frac{(m-1)^2}{2} \rfloor \frac{2\pi}{m}]$. In this case, increasing m results in an approximately linear increase in the required phase shift $\Delta\Phi_{max}$ and the phase shifter length.

The unit phase shifter length L that creates $\frac{2\pi}{m}$ phase shift is determined by the modulation efficiency ($V_{\pi \cdot cm}$) of the phase shifter material and the maximum bias voltage (V_{max}). Essentially, low $V_{\pi \cdot cm}$ and high V_{max} results in a short device length.

For high-speed modulation in silicon photonics phase shifters with bandwidths ≥ 1 GHz, the most commonly used actuation mechanisms rely on plasma dispersion to control the refractive index by manipulating the free carrier density. For such phase shifters, prior work demonstrated $V_{\pi \cdot cm}$ values lower than 0.5 V.cm [7-11] and optical losses less than 1 dB/cm [12-13]. Fig. R1 shows the total phase shifter length for a multiplication unit (L_{mul}) versus m for various high-bandwidth (≥ 10 GHz) phase shifter implementations [7-11].

Fig. R1: Device length for a single multiplication unit (L_{mul}) versus modulus value (m). The plot includes devices based on free carrier dispersion via carrier depletion [7], accumulation [8], and injection [9] in silicon, as well as hybrid shifters integrated with InGaAs [10] and organic materials [11].

To determine the total length in this RNS-based approach, the moduli choice for a required RNS range is critical. To minimize the device length, one can pick moduli as small as possible. This enables using a shorter total phase shifter length, however, utilizes more MVM arrays in the iso-throughput case.

Fundamentally, given the device characteristics, precision requirements of the DNN workloads, and a hardware budget, it is essential to conduct a thorough design space exploration to optimize performance, power, and area of the overall accelerator—which is beyond the scope of this paper. However, to quantify with a specific example, let us consider the following co-prime moduli set achieving a dynamic range of more than 17 bits—which allows a vector size as large as 90 when 6-bit input/weight data are used: {5, 7, 8, 9, 11, 13}. In this case, the longest device length for the selected moduli set varies between 0.3-1.2 mm for the best case (0.032 V_{cm} @ 2.8V) and 8.3-28.8 mm for the worst case (0.52 V_{cm} @ 2V) in Fig. R1, respectively. With a typical device width of 25 μm and a chip size of 500 mm², an array size of 64 × 64 (six arrays in total, one 64 × 64 array for each modulus in the abovementioned moduli set) can be achieved.

It is expected that the proposed RNS-based design to be less area efficient and have higher optical loss per MAC operation compared to a traditional MZI due to the relatively long phase shifter lengths and utilization of multiple MVM arrays. However, this approach is feasible and it enables the precision in the optical channel to be reduced to $\log_2 m \leq 4$ -bit (for the example above), which can tolerate higher optical loss, while achieving a ~17-bit output. An equivalent precision in traditional photonic cores requires 2¹⁷ differentiable analog levels at the output of the optical MAC operations and 17-bit ADCs, which is impractical with today's technology.

The scalability of the RNS-based approach can be further improved with the developments in photonic technology. Developing high-bandwidth phase shifters with low $V_{\pi/cm}$ and low optical loss is still an active research area. Integration of new materials (e.g. III–V semiconductors[10], organic materials[11], ferroelectrics [12,16], (silicon-)germanium [14], and 2D materials [15]) provides promising results despite still being in very early stages. With these integration technologies maturing, more performant phase shifters can enable better area efficiency. In addition, vertically stacking up multiple photonic chiplets via 3D integration (e.g., photonic arrays for different moduli can be implemented on different layers) can further increase the computational density in such designs.

Action taken: We added details about the device length and scalability of the RNS-based photonic design in Section “Modular Arithmetic with Phase Shifters” on page 10 under “Methods” (paragraphs 3-6):

“In this approach, the total length of the phase shifter on each arm depends on m and the vector size h . Therefore, achieving a feasible design requires careful moduli set and device selection. During an RNS multiplication with modulus m where both x and w are smaller than m , the maximum multiplication result is $(m - 1)^2$ which can be mapped around zero as $[-\lfloor \frac{(m-1)^2}{2} \rfloor, \lfloor \frac{(m-1)^2}{2} \rfloor]$. For a modular dot product unit with h elements, the range of the phase shift that the unit can introduce must be $[-\Phi_{max}, \Phi_{max}] = [-\lfloor \frac{(m-1)^2}{2} \rfloor \frac{2\pi}{m} h, \lfloor \frac{(m-1)^2}{2} \rfloor \frac{2\pi}{m} h]$ when the maximum bias voltage V_{max} is applied. This requires a total phase shifter length of $O(mh)$ in the dot product unit.

Here, the unit phase shifter length L that creates $\frac{2\pi}{m}$ phase shift is determined by the modulation efficiency ($V_{\pi\text{-cm}}$) of the phase shifter material and the maximum bias voltage (V_{max}). Essentially, a low $V_{\pi\text{-cm}}$ and high V_{max} results in a short device length for the required phase shift. For high-speed phase shifters with modulation bandwidths ≥ 1 GHz, the most commonly used actuation mechanisms rely on plasma dispersion to control the refractive index by manipulating the free carrier density. For such phase shifters, prior work demonstrated $V_{\pi\text{-cm}}$ values lower than 0.5 V.cm and optical losses less than 1 dB/cm.

To determine the total length in this RNS-based approach, the required RNS range (depends on the input precision and vector size) and the corresponding moduli choice are critical. A moduli set with fewer but larger values requires fewer but longer dot product units, while a moduli set with more but smaller moduli results in many but shorter dot product units.

To quantify, an example moduli set {5, 7, 8, 9, 11, 13} can achieve a dynamic range of more than 17 bits—which allows 6-bit arithmetic up to $h = 90$. When a phase shifter with 0.032 V.cm modulation efficiency at 2.8 V is used, the phase shifter length varies between 0.3 mm to 1.2 mm (per multiplication) for different moduli. With a typical device width of 25 μm , an array size of 64×64 (six arrays in total, one 64×64 array for each modulus in the abovementioned moduli set) can fit in a typical chip size of 500 mm^2 . This approach is less area efficient and has higher optical loss per MAC operation compared to a traditional MZI array due to the relatively long phase shifter lengths and utilization of multiple MVM arrays. However, this approach is feasible and it allows us to use lower-precision optical channels (2-to-4-bit for the example above), which can tolerate higher optical loss while achieving a ~ 17 -bit output. An equivalent precision in traditional photonic cores requires 2^{17} differentiable analog levels at the output of the optical MAC operations and 17-bit ADCs, which is impractical with today's technology (See Fig. 6a).

The scalability of the RNS-based approach can be further improved with the developments in photonic technology. Developing high-bandwidth phase shifters with low $V_{\pi\text{-cm}}$ and low optical loss is still an active research area. Integration of new materials (e.g. (silicon-)germanium, ferroelectrics, III–V semiconductors, 2D materials, and organic materials) provides promising results despite still being in very early stages. With these integration technologies maturing, more performant silicon photonics phase shifters can enable better area efficiency. In addition, using 3D integration to stack up photonic chiplelets (e.g., photonic arrays for different moduli can be implemented on different layers) can further reduce the area footprint in such designs.”

Reviewer’s comment: “Furthermore, the selection of ‘w’ and ‘v’ seems somewhat counter-intuitive. Typically, ‘w’ is pre-trained, while ‘x’ (or ‘v’) is often considered variable. However, the authors treat ‘w’ as a pure digital number, which results in an increased number and physical size of phase shifters. In contrast, ‘x’ is entirely analog. The paper should delve into a more comprehensive discussion regarding the rationale behind this approach.”

Response: We agree with the reviewer that for some neural networks, w is pre-trained and fixed and x is computed at runtime (where w represents the model parameters and x represents the input to those parameters). Ideally, the pre-trained w can be programmed onto the photonic

devices once and kept fixed for multiple inferences. However, today's DNN with millions to billions of parameters makes it impossible to map a whole DNN onto a single accelerator. Therefore, although DNN parameters (w) are not calculated during runtime, they have to be tiled into smaller pieces and loaded into the photonic devices tile by tile. Additionally, modern neural networks that use attention modules require multiplications between matrices that cannot be pre-computed, i.e., $Attention(Q, K, V) = Softmax(Q @ KT / \sqrt{d}) @ V$, where $@$ denotes matrix-multiplication and Q , K , and V must be computed during runtime. To this end, both x and w are stored as digital values in the memory before the operations. In our design, one variable is used as a digital value to control the phase shifters digit-by-digit. The second variable is converted into the analog domain to represent an analog voltage value. As both x and w are digital values in the beginning and they have the same bit-width, the order of these variables can be easily exchanged, i.e., x can be used digit-by-digit and w can be used as an analog value or vice versa.

Action taken: We added the following paragraph in the Subsection "Modular Arithmetic with Phase Shifters" on page 9 under the Methods section:

"In the example in Fig. 7b w is a digital number encoded digit-by-digit to control the phase shifters separately, while x is encoded via an analog voltage v . Ideally, the pre-trained w (for inference) can be programmed onto the photonic devices once and kept fixed for multiple inferences. However, today's DNNs contain millions to billions of parameters, which makes it impossible to map a whole DNN onto a single accelerator. Therefore, although DNN parameters are not calculated during runtime, w has to be tiled into smaller pieces and loaded into the photonic devices tile by tile. Additionally, modern neural networks that use attention modules require multiplications between matrices that cannot be pre-computed. As a result, both x and w are stored as digital values in the memory before the operations. To this end, the order of these variables can be easily exchanged, i.e., x can be programmed digit-by-digit and w can be used as an analog value or vice versa. "

Reviewer #2:

Reviewer's comment: "More clarification is required to differentiate the proposed approach from the prior works. Similar approaches for AI acceleration and digital-analog data conversion are found in "Salamat et al, RNSnet: In-Memory Neural Network Acceleration Using Residue Number System, IEEE International Conference on Rebooting Computing (ICRC) 2018" and "J. Peng, et al, DNNARA: A Deep Neural Network Accelerator using Residue Arithmetic and Integrated Photonics. ICPP: 61:1-61:11 (2020)"."

Response: RNS is a well-explored numeral system, which has been used in a variety of applications including digital signal processing [17], cryptography [18], and DNNs [1, 2, 3]. RNS-based DNN computation in digital hardware was proposed for improving energy efficiency by breaking numbers into residues with fewer bits. RES-DNN [1] implements an RNS-based version of the popular DNN accelerator Eyeriss and RNS-Net [2] uses a processing-in-memory (PIM)-based design and simplifies RNS operations to PIM-friendly ones. A similar work, DNNARA [3], is a nanophotonic (not analog) RNS-based DNN inference accelerator where the

authors use 2x2 optical switches to manipulate the route of the light to perform multiplication and additions using a one-hot encoded mapping.

While all three works are similar to our study in terms of using RNS for DNN inference, we are the first to propose using RNS in the context of analog DNN computation. Moreover, we believe that the motivation and execution of our study are fundamentally different from the previous works. Our work aims to resolve the limited precision issue inherent in analog hardware and enable the effective use of analog compute cores in DNN inference and training by achieving high precision and high DNN accuracy through RNS.

Different from our approach, these accelerators all employ fully RNS-based dataflows without switching back and forth between RNS and BNS. Although this approach removes the cost of RNS-BNS conversions, it requires periodically performing overflow detection and RNS ranging operations in the RNS domain to preserve the integrity of RNS operations. In addition, these fully RNS-based designs force the end-to-end DNN to be computed in fixed-point arithmetic. Performing nonlinear operations in the RNS domain requires using approximations (e.g., Taylor series expansion) to reduce nonlinear operations into multiply and add operations. These approximations in nonlinear functions cause information loss and demand higher data precision. As a result, these works use 16-bit or higher precision to represent data to achieve high accuracy and their designs are limited to DNN inference. Our study aims to reduce the required data precision in analog hardware as much as possible while still maintaining high DNN accuracy. To this end, switching back and forth between RNS and BNS for each matrix-vector multiplication (MVM) operation allows us to control the precision of nonlinear operations (which are performed on digital hardware) independently and perform scaling before MVM operations to alleviate the quantization errors at the data converters (See Accuracy Modeling under Methods on page 8). This approach also enables us to perform backpropagation and successfully train DNNs with low-precision arithmetic besides DNN inference. We demonstrate successful DNN inference with 6-bit and training with 7-bit data precision in analog hardware. Lastly, different from previous works, we analyze the impact of noise in RNS-based DNN inference and integrate RRNS to combat the accuracy loss caused by the errors in analog hardware.

Action taken: We added a related work paragraph in the Discussion on page 8. The paragraph is copied below for easy reference:

“RNS is a well-explored numeral system, which has been used in a variety of applications including digital signal processing, cryptography, and DNNs. RNS-based DNN computation in digital hardware was proposed for improving energy efficiency by breaking numbers into residues with fewer bits. Res-DNN proposes an RNS-based version of the popular DNN accelerator Eyeriss and RNS-Net uses a processing-in-memory (PIM)-based design and simplifies RNS operations to PIM-friendly ones. A similar work, DNNARA, is a nanophotonic (not analog) RNS-based DNN inference accelerator where the authors use 2×2 optical switches to manipulate the route of the light to perform multiplication and additions using a one-hot encoded mapping. While all three works are similar to our study in terms of using RNS for DNN inference, we are the first to propose using RNS in the context of analog DNN computation. In addition,

these accelerators all propose fully RNS-based dataflows without switching back and forth between RNS and BNS. Although this approach of staying in the RNS domain removes the cost of RNS-BNS conversions, it requires periodically performing overflow detection and ranging operations in the RNS domain to preserve the integrity of RNS operations. More importantly, these fully RNS-based computations force the end-to-end DNN to be computed in fixed-point arithmetic. Performing nonlinear operations in the RNS domain requires using approximations (e.g., Taylor series expansion) to reduce nonlinear operations into multiply and add operations. These approximations in nonlinear functions cause information loss and demand higher data precision. As a result, these previous works use 16-bit or higher precision to represent data to achieve high accuracy and their proposals are limited to DNN inference. In our approach, switching back and forth between RNS and BNS for each MVM operation allows us to control the precision of nonlinear operations (which are performed on digital hardware) independently and perform scaling (dynamic quantization) before MVM operations to alleviate the quantization errors at the data converters (See Accuracy Modeling under Methods). This approach also enables us to perform backpropagation and successfully train DNNs with low-precision arithmetic besides DNN inference (inference with 6-bit and training with 7-bit). In contrast to the few previous analog DNN training demonstrations that were limited to very simple tasks (e.g., MNIST classification) and DNNs with a few small layers, our approach can achieve a much higher dynamic range through RNS and can successfully train state-of-the-art DNNs. Lastly, different from previous works, we analyze the impact of noise in RNS-based DNN inference and integrate RRNS to combat the accuracy loss caused by the errors in analog hardware.”

Reviewer’s comment: “What’s new about the RRNS algorithms in this work if they are already developed and used in Ref. 25-27? A detailed comparison to prior works has to be provided.”

Response: In our work, we use the RRNS algorithm as described in the works mentioned by the reviewer (Ref. 25-27). This involves adding multiple redundant moduli to the set and performing operations for all the moduli (redundant and non-redundant) in the set. Then, the error correction is done through major logic decoding. Ref. 25-27 propose the theory of RRNS and investigate its impact on error distributions in RNS operations in the context of channel communication. Our contribution here is not the error correction methodology, but the proof that the fault tolerance capability of RRNS can maintain high DNN accuracy in analog hardware in the presence of noise.

Reviewer’s comment: “What clock rate is proposed in the calculation of energy consumption? How is k_1 and k_2 calculated in Eq. 11? Is there energy overhead in operating multiple low-speed ADCs?”

Response: The energy consumption of ADCs was calculated using Eq. (11) stated in the Methods section of the original manuscript. This equation, i.e., $E_{ADC}(b) = k_1 b + k_2 4^b$, maps the bit precision (b) of the ADC to its energy consumption per conversion (E_{ADC}) and is independent of the clock frequency, technology node, ADC type, etc. In the first manuscript, we used this equation from Murmann’s study in 2020 [19] with their k_1 and k_2 values estimated using the data available in Murmann’s ADC survey [20]. This survey includes all the literature on

ADCs published since 1997 in the International Solid-State Circuits Conference (ISSCC) and the VLSI Circuit Symposium—the two main venues in the field.

After the reviewer’s comment, we removed data points with low sampling frequency ($f_s \leq 1\text{GHz}$) as our design requires high-speed data converters, added the most recent data from the same ADC survey to our analysis, and recalculated the coefficients, k_1 and k_2 , according to the new dataset using the same methodology in [9]. k_1 is calculated as the average of the three lowest E_{ADC}/b and k_2 is calculated as the average of the three lowest $E_{ADC}/4^b$ within the samples of the dataset. The up-to-date dataset with $f_s > 1\text{ GHz}$ and the recalculated energy function are shown in Fig. R2. The fit resulted in $k_1 = 0.11\text{ pJ}$ and $k_2 = 0.241\text{ aJ}$.

Fig. R2: ADC data and the updated energy function. The data includes all literature on ADCs published in ISSCC and VLSI conferences since 1997.

Actions taken: We added Fig. R2 along with details about the ADC energy estimation with the updated dataset to the manuscript in Section “Data Converter Energy Estimation” under “Methods” (pages 8-9). The added paragraph is copied below:

“For calculating the coefficients k_1 and k_2 , we used the data from the ADC survey collected by Murmann. Fig. 7 shows the ADC data points from the survey and the fitting function. The data includes all the ADC literature published in the two main venues of the field, the International Solid-State Circuits Conference (ISSCC) and the VLSI Circuit Symposium, between the years 1997 and 2023. We removed the data points with a sampling frequency (f_s) lower than 1 GHz as our design requires high-speed data converters. k_1 is calculated as the average of the three minimum E_{ADC}/b and k_2 as the average of the three minimum $E_{ADC}/4^b$ among the data points.”

We also updated the energy plot in Fig. 6a in the updated manuscript (it was Fig. 2b in the first manuscript) using the new coefficients in the ADC energy equation. While the energy gains have decreased compared to the first version of the plot (by less than an order of magnitude), the message remains the same as the exponential relationship between bit precision and energy consumption still exists. Fig. R3 shows the updated plot for easy reference:

Fig. R3: Updated energy plot (Fig. 6a in the updated manuscript)

Reviewer’s comment: “In the redundant RNS, what’s the cost (energy or compute time) of using the k redundant moduli to make the system fault tolerant?”

Response: With redundant RNS, the same operations are performed on the $n + k$ moduli in total (k redundant and n non-redundant) instead of n . This results in an approximately linear increase in energy consumption with the increasing number of moduli. The energy consumption of DACs and ADCs for different k is shown in Fig. R4. The energy consumption of the RRNS is shown for 5-8 bits as there are not enough co-prime moduli smaller than 15 to use RRNS for the 4-bit case.

The compute time does not increase with the increase in k as operations for different moduli are independent and can be performed in parallel. BNS-to-RNS conversion, D-to-A conversion, analog MVM, A-to-D conversion, RNS-to-BNS conversion, and error detection steps can be pipelined in the dataflow. Therefore, adding error correction to the dataflow adds another step to the pipeline (independent from k), but does not impact the throughput.

Fig. R4: Data converter energy consumption per dot product for RNS ($k=0$) and RRNS ($k=1,2,4$)

Action taken: We moved the “Energy Efficiency in the RNS-based Analog Core” subsection to the end of the “Results” section to include the energy consumption of RRNS in the same section which we titled “Energy and Area Efficiency in the RNS-based Analog Core”. We also added the plot in Fig. R4 showing the energy consumption for different k values (Fig. 6b in the updated manuscript) and the associated discussion shown below in Section “Energy Efficiency in the RNS-based Analog Core” on pages 6-7:

“Fig. 6b shows the energy consumption of DACs and ADCs when RRNS is used. RNS results in an approximately linear increase in energy consumption with the increase in the number of moduli ($n + k$). The plot only shows the 5-to-8-bit cases as there are not enough co-prime moduli smaller than 15 to use RRNS for the 4-bit case. The compute time does not increase with the increase in k as operations for different moduli are independent and can be performed in parallel.”

Reviewer #3:

Comment: “The authors only compare the energy consumption and accuracy of different approaches. But it would also be helpful to analyze how much area it takes using RNS compared to conventional HP and LP ADC approach (at iso-throughput setting), i.e., would having n -copy of DAC/ADC take up a significant area of the chip?”

Response: The area footprint of DACs and ADCs has a weaker relationship with bit precision compared to their energy consumption. In a study by Verhelst and Murmann in 2012 [21], the authors observed that the area footprint of ADCs is proportional to $2^\alpha b$ where $\alpha \in [0.11, 1.07]$ depending on the type of the ADC, and $\alpha = 0.5$ when all types are considered. Assuming the same technology node is used for all the ADCs and $\alpha = 0.5$, Fig. R5 compares the normalized

area footprint of ADCs for the low- and high-precision (LP and HP) fixed-point, RNS ($k = 0, n$ ADCs per dot product) and RRNS-based ($k > 0, n + k$ ADCs per dot product) approaches. While the area footprint of the RNS and RRNS-based approaches are 3-10 \times higher than the LP fixed-point approach, they have a more than 10 \times smaller area footprint than the HP fixed-point approach for all bit precisions. In addition, the same study points out that the sampling frequency of ADCs is independent of the area footprint. Therefore, in the RNS and RRNS approaches, instead of having multiple ADCs per dot product, one can use a single and faster ADC and perform multiple conversions using the same ADC to achieve the same throughput with better area efficiency.

Fig. R5: Normalized area footprint of ADCs for the LP and HP fixed-point, the RNS ($k = 0, n$ ADCs per dot product) and RRNS-based ($k > 0, n + k$ ADCs per dot product) approaches.

Action taken: We added Fig. R5 (Fig. 6c in the updated manuscript) and the following paragraph to the Section “Energy and Area Efficiency in the RNS-based Analog Core” on page 6-7:

“The area footprint of data converters has a weaker correlation with bit precision than their energy consumption. In a study by Verhelst and Murmann in 2012, the authors observed that the area footprint of ADCs is proportional to $2^{\alpha b}$ where $\alpha \in [0.11, 1.07]$ depending on the type of the ADC, and is $\alpha = 0.5$ when all ADC types are considered. Assuming the same technology node is used, Fig. 6c shows the normalized area footprint of ADCs for the LP and HP fixed-point, RNS ($k = 0, n$ ADCs per dot product), and RRNS-based ($k > 0, n + k$ ADCs per dot product) approaches. While the area footprint of the RNS and RRNS-based approaches are higher than the LP fixed-point approach, they have a smaller area footprint than the HP fixed-point approach for all bit precisions. In addition, the same study points out that the sampling frequency of ADCs is independent of the area footprint. Therefore, in the RNS and RRNS approaches, instead of having multiple ADCs per dot product, one can use a single and

faster ADC and perform multiple conversions using the same ADC to achieve the same throughput with better area efficiency.”

Reviewer’s comment: *“The authors only analyze the accuracy impact of the proposed error detection/correction scheme using redundant RNS. It would also be interesting to see how much energy and area overhead the redundant RNS takes.”*

Response: With redundant RNS, all the operations are performed on the $n + k$ moduli in total (k redundant and n non-redundant) instead of n . This results in an approximately linear increase in energy consumption with the increasing number of moduli. The energy consumption of DACs and ADCs for different k is shown in Fig. R4. The energy consumption of the RRNS is shown for 5-8 bits as there are not enough co-prime moduli smaller than 15 to use RRNS for the 4-bit case.

Action taken: We moved the “Energy Efficiency in the RNS-based Analog Core” subsection to the end of the “Results” section to include the energy consumption of RRNS in the same section which we titled “Energy and Area Efficiency in the RNS-based Analog Core”. We also added the plot in Fig. R4 showing the energy consumption for different k values (Fig. 6b in the updated manuscript) and the associated discussion shown below in Section “Energy Efficiency in the RNS-based Analog Core” on pages 6-7:

“Fig. 6b shows the energy consumption of DACs and ADCs when RRNS is used. RNS results in an approximately linear increase in energy consumption with the increase in the number of moduli ($n + k$). The plot only shows the 5-to-8-bit cases as there are not enough co-prime moduli smaller than 15 to use RRNS for the 4-bit case. The compute time does not increase with the increase in k as operations for different moduli are independent and can be performed in parallel.”

Reviewer’s comment: *“The RNS needs additional modular arithmetic, but the cost of this part is not presented in the paper. It would be interesting to see how much additional energy and area the modular arithmetic takes.”*

Response:

The operations in RNS involve BNS-to-RNS (forward) and RNS-to-BNS (reverse) conversions in the digital domain and modular multiply-accumulate (MAC) operations in the analog domain. The cost of the forward and reverse conversions is dependent on the chosen moduli set and the conversion technique. Reverse conversion can be performed via CRT, mixed radix conversion (MRC), or lookup tables. In addition, special moduli sets and hardware implementations can alleviate the cost of these conversions by reducing modulo operations into simpler operations (e.g., shift operations for moduli in the form of powers of two). Prior work shows that these circuits can be implemented with \leq mW power consumption, \geq 2 GHz throughput, and \leq 0.0015 mm² area in 65 nm CMOS for a dynamic range larger than 23 bits (\sim 0.5 pJ/conversion) [22].

The modular MAC operations require performing n MAC operations for n moduli and a following modulo operation in the analog domain. The energy/area costs on the data converters

(DACs/ADCs) in the Section “Energy and Area Efficiency in the RNS-based Analog Core” as we propose a technology-agnostic approach and the data conversion has been pointed out to be the major bottleneck in most analog systems. One can conservatively assume that the energy consumption and area for the analog operations will increase by n times compared to a traditional analog hardware. However, the exact cost of analog operations is highly dependent on can be optimized for the accelerator design, device characteristics, and chosen analog technology.

For example, the photonic approach with multiple phase shifters explained in Section “Discussion” integrates the analog modulo operations with the photonic dot products. Therefore, the analog modulo does not cause any additional energy overhead. This design, however, results in longer phase shifters and higher optical loss compared to a typical MZI, but also reduces the bit precision required in the optical channels and, in turn, can tolerate higher optical loss. Please see the highlighted text in the Section “Modular Arithmetic with Phase Shifters” for details (also copied below as part of the taken actions).

Alternatively, ring oscillators can simply be added after any analog MAC operation that outputs an analog voltage or current. Recent works show that it is possible to achieve tens of GHz of oscillation frequency with energy consumption as low as 20 aJ for a 7-stage ring oscillator [23]. The total energy overhead of the ring oscillator depends on the total number of steps (output of a dot product, i.e., $y = \sum_{j=1}^h x_j w_j$) and the modulus m . The energy consumption of the modulo operation is then $y * 20/7$ aJ where $y = (m - 1)^2 h$ in the worst-case scenario. ≤ 1 pJ per modulo energy budget can easily achieve an RNS range of more than 18 bits. The ring oscillator consists of m inverters for mod m operation, therefore, the area increases linearly with m and depends on the transistor technology, however, is typically quite small compared to other components such as ADCs.

Action taken: We added a paragraph in “Discussion” about the cost of the RNS-BNS conversion circuits (paragraph 2). The paragraph is highlighted in blue in the manuscript and copied over below for easy reference:

“Using RNS requires forward and reverse conversion circuits to switch between the RNS and the binary number system (BNS). The forward conversion is a modulo operation while the reverse conversion can be done using the CRT, mixed-radix conversion, or look-up tables. The (digital) hardware costs of these circuits can be reduced by choosing special moduli sets. Prior work has shown that it is possible to implement these conversions ≤ 1 mW power, ≥ 2 GHz throughput and ≤ 0.0015 mm² area in 65 nm CMOS for a dynamic range larger than 23 bits (≤ 0.5 pJ/conversion).”

We added details about the phase shifter length and area in the phase shifter-based modular arithmetic approach in the Section “Modular Arithmetic with Phase Shifters” under “Methods” starting from paragraph 3 on page 10-11. The text is highlighted in blue in the manuscript and copied over below for easy reference:

“In this approach, the total length of the phase shifter on each arm depends on m and the vector size h . Therefore, achieving a feasible design requires careful moduli set and device selection. During an RNS multiplication with modulus m where both x and w are smaller than m , the maximum multiplication result is $(m - 1)^2$ which can be mapped around zero as $[-\lfloor \frac{(m-1)^2}{2} \rfloor, \lfloor \frac{(m-1)^2}{2} \rfloor]$. For a modular dot product unit with h elements, the range of the phase shift that the unit can introduce must be $[-\Phi_{max}, \Phi_{max}] = [-\lfloor \frac{(m-1)^2}{2} \rfloor \frac{2\pi}{m} h, \lfloor \frac{(m-1)^2}{2} \rfloor \frac{2\pi}{m} h]$ when the maximum bias voltage V_{max} is applied. This requires a total phase shifter length of $O(mh)$ in the dot product unit.

Here, the unit phase shifter length L that creates $\frac{2\pi}{m}$ phase shift is determined by the modulation efficiency ($V_{\pi\text{-cm}}$) of the phase shifter material and the maximum bias voltage (V_{max}). Essentially, a low $V_{\pi\text{-cm}}$ and high V_{max} results in a short device length for the required phase shift. For high-speed phase shifters with modulation bandwidths ≥ 1 GHz, the most commonly used actuation mechanisms rely on plasma dispersion to control the refractive index by manipulating the free carrier density. For such phase shifters, prior work demonstrated $V_{\pi\text{-cm}}$ values lower than 0.5 V.cm and optical losses less than 1 dB/cm.

To determine the total length in this RNS-based approach, the required RNS range (depends on the input precision and vector size) and the corresponding moduli choice are critical. A moduli set with fewer but larger values requires fewer but longer dot product units, while a moduli set with more but smaller moduli results in many but shorter dot product units.

To quantify, an example moduli set $\{5, 7, 8, 9, 11, 13\}$ can achieve a dynamic range of more than 17 bits—which allows 6-bit arithmetic up to $h = 90$. When a phase shifter with 0.032 V.cm modulation efficiency at 2.8 V is used, the phase shifter length varies between 0.3 mm to 1.2 mm (per multiplication) for different moduli. With a typical device width of 25 μm , an array size of 64×64 (six arrays in total, one 64×64 array for each modulus in the abovementioned moduli set) can fit in a typical chip size of 500 mm^2 . This approach is less area efficient and has higher optical loss per MAC operation compared to a traditional MZI array due to the relatively long phase shifter lengths and utilization of multiple MVM arrays. However, this approach is feasible and it allows us to use lower-precision optical channels (2-to-4-bit for the example above), which can tolerate higher optical loss while achieving a ~ 17 -bit output. An equivalent precision in traditional photonic cores requires 2^{17} differentiable analog levels at the output of the optical MAC operations and 17-bit ADCs, which is impractical with today’s technology (See Fig. 6a).

The scalability of the RNS-based approach can be further improved with the developments in photonic technology. Developing high-bandwidth phase shifters with low $V_{\pi\text{-cm}}$ and low optical loss is still an active research area. Integration of new materials (e.g. (silicon-)germanium, ferroelectrics, III–V semiconductors, 2D materials, and organic materials) provides promising results despite still being in very early stages. With these integration technologies maturing, more performant silicon photonics phase shifters can enable better area efficiency. In addition,

using 3D integration to stack up photonic chiplets (e.g., photonic arrays for different moduli can be implemented on different layers) can further reduce the area footprint in such designs.”

We added the cost of analog modulo operation using ring oscillators in the Section “Modular Arithmetic with Ring Oscillators” under “Methods” starting in paragraph 3 on page 10. The text is highlighted in blue in the manuscript and copied over below for easy reference:

“In analog hardware, dot products can be performed using traditional methods with no change and with any desired analog technology where the output can be represented as an analog electrical signal (e.g., current or voltage) before the analog modulo. The ring oscillator is added to the hardware where the dividend A is the output of the dot product. Here, the total energy consumption of the analog modulo operation depends on A . Recent works show that it is possible to achieve tens of GHz of oscillation frequency with energy consumption lower than 20 aJ for a 7-stage ring oscillator. In the worst-case scenario where $A = (m - 1)^2 h$ in an h -long dot product, a modulo operation can be performed with ≤ 1 pJ energy consumption for a moduli set can achieve a dynamic range of more than 18 bits. The ring oscillator consists of m inverters, which typically has a quite smaller area footprint than the other components such as ADCs.”

Reviewer’s comment: *“The $\geq 99\%$ of FP32 accuracy claimed in the paper is achieved without adding noise to the system. It is not discussed in the paper how much accuracy their proposed system can achieve under the typical noise level in an analog circuit.”*

Response: The noise level in analog circuits depends on the underlying analog technology, device characteristics, and many other factors. An accurate noise estimation requires focusing on a single technology and accelerator design, and conducting detailed simulations or taking measurements from real hardware, which is beyond this paper’s scope.

As an example, we can consider a mathematical model for a photonics-based analog RNS accelerator design limited by shot noise and thermal noise. Here, we model both noises as Gaussian distributions that are additive to the output value, i.e., $\sum_j x_j w_j + \mathcal{N}(0, 1) \sigma_{noise}$ for a dot

product. Here, $I_{shot} \sim \sqrt{2q_e \Delta f I_{out}} \mathcal{N}(0, 1)$ and $I_{th} \sim \sqrt{\frac{4k_B T \Delta f}{R_{TIA}}} \mathcal{N}(0, 1)$, where q_e is the elementary charge, Δf is the bandwidth, I_{out} is the output current of the analog dot product, k_B is the Boltzmann constant, T is the temperature, and R_{TIA} is the feedback resistor of the transimpedance circuitry.

During operations with integers $\in [0, m - 1]$ for modulus m , two consecutive output residues must be I_{out}/m apart from each other when scaled between $[0, I_{out}]$ for a maximum analog output current I_{out} to be able to differentiate m distinct levels. If the noise gets larger than $I_{out}/2m$ for an output residue, it is rounded to the next integer. Therefore, the error probability of

a single residue is $p = \mathbb{P}(\sqrt{I_{shot}^2 + I_{th}^2} \geq \frac{I_{out}}{2m})$. For RNS without any redundant moduli, this

results in an error at the binary output with a probability of $p_{err} = 1 - (1 - p)^n$ for n moduli. For RRNS, p_{err} for a given p can be obtained by using Fig. 4 in the manuscript.

Below, we provide p_{err} versus I_{out} for RNS with 6, 7, and 8-bit moduli in Fig. R6 (the moduli sets for different bit precisions are provided in Table I in the manuscript). We also show the cut-off p_{err} values we identified in our experiments for ResNet50 and BERT-Large (See Fig. 5(a)-(f) in Section “Redundant RNS for Fault Tolerance” in the manuscript). This plot shows that p_{err} decreases with the increasing I_{out} and decreasing m . For example, for the 6-bit case, less than 1 mA output current is adequate to prevent accuracy loss in both DNNs. This corresponds to a ~1mW (0 dBm) output power at the photodetector where the photodetector responsivity is ~1 A/W—which is feasible assuming a typical 10 dBm laser source and 10 dB loss along the optical path. Effectively, increasing the input power leads to higher output current and can maintain high accuracy for higher p , which creates a tradeoff between the input power and noise tolerance in analog accelerators.

Fig. R6: p_{err} versus I_{out} for RNS ($k=0$) and RRNS ($k=1$) with 6, 7, and 8-bit moduli. RRNS case is for a single attempt of error correction.

Alternatively, as proposed in Section “Redundant RNS for Fault Tolerance”, redundant moduli can be used for fault tolerance. Fig. R6 also shows p_{err} versus I_{out} for RRNS with 6, 7, and 8-bit moduli when one redundant modulus is used for a single attempt of error correction (cyan lines). Using redundant moduli reduces the required I_{out} so it can avoid accuracy loss for lower I_{out} . The added redundant modulus to the 6-bit case reduces the required I_{out} down to 0.5 mA as against 1 mA when $k=0$.

Action taken: We modified Fig. 5 to include p_{err} versus I_{out} plots for RRNS for different bit precisions, numbers of moduli, and numbers of attempts. We added the following paragraph to the Section “Redundant RNS for Fault Tolerance”:

“In analog hardware, expected p and p_{err} depend on the underlying analog technology, device characteristics, and many other factors. As an example, we study a photonics-based RNS analog accelerator design that is thermal and shot noise-limited. The noise can be modeled as a Gaussian distribution that is additive to the output value, i.e., $\sum_j x_j w_j + \mathcal{N}(0, 1)\sigma_{noise}$ for a dot product. Many other noise sources present in other analog technologies can be represented using a similar framework. For an analog core where output is encoded in an analog current, let us define the maximum achievable current as I_{out} . A higher I_{out} results in a higher SNR and lower p_{err} , creating a tradeoff between input power and noise tolerance.

Without any redundant moduli ($k = 0$), $I_{out} \leq 1$ mA is adequate to prevent accuracy loss in both DNNs (the cut-off is at 2 mA and 8 mA for 7-bit and 8-bit cases, respectively). For instance, for a photonic system, this requires ~ 1 mW (0 dBm) output power (for a photodetector with ~ 1 A/W responsivity)—which is feasible assuming a 10 dBm laser source and 10 dB loss along the optical path.

The required I_{out} can be further lowered by using RRNS. Fig.5 (g-i) shows the relationship between I_{out} and the expected p_{err} for different RRNS. For a smaller number of bits and a higher k , a lower I_{out} is required to stay under the cut-off p_{err} for the evaluated DNNs. For example, a 6-bit RRNS with $k = 1$ requires 0.1 mA for a single error correction attempt as against the $k = 0$ where 1 mA is needed to avoid accuracy loss due to analog noise. The required I_{out} similarly decreases with the increasing number of attempts. Please see Noise Modeling under Methods for details.”

We added the subsection “Noise Analysis” under “Methods” to include the mathematical noise modeling. The added text is copied below and highlighted in the manuscript for ease of reference:

“In analog hardware, both shot noise and thermal noise can be modeled as Gaussian distributions, i.e., $I_{shot} \sim \sqrt{2q_e \Delta f I_{out}} \mathcal{N}(0, 1)$ where q_e is the elementary charge, Δf is the bandwidth, I_{out} is the output current of the analog dot product and $I_{th} \sim \sqrt{\frac{4k_B T \Delta f}{R_{TIA}}} \mathcal{N}(0, 1)$ where k_B is the Boltzmann constant, T is the temperature, and R_{TIA} is the feedback resistor of the transimpedance circuitry.

For an RNS operation with modulus m the output residues should be at least I_{out}/m apart from each other to differentiate m distinct levels. An error occurs in the output residue when

$\sqrt{I_{shot}^2 + I_{th}^2} \geq \frac{I_{out}}{2m}$ as the residue is rounded to the next integer. Therefore, the error probability in a single residue can be calculated as $p = \mathbb{P}(\sqrt{I_{shot}^2 + I_{th}^2} \geq \frac{I_{out}}{2m})$.

We used $\Delta f = 5$ GHz, $T = 300$ K, and $R_{TIA} = 200 \Omega$ as typical values in the experiments shown in Fig. 5(g-i). For a given p , $p_{err} = 1 - (1 - p)^n$ for and RNS ($k = 0$) with n moduli. For RRNS ($k > 0$), p_{err} can be obtained using Fig. 4 or Eq. (8)."

Conclusion

In conclusion, we have discussed above how our work tackles the precision limitation in analog computing and enables the usage of analog hardware in high-precision tasks such as DNN training using RNS. We thank the referees again for their comments, and we hope that our changes to the manuscript help clarify the impact of our work.

REFERENCES

- [1] Samimi, N., Kamal, M., Afzali-Kusha, A., & Pedram, M. Res-DNN: A residue number system-based DNN accelerator unit. IEEE Transactions on Circuits and Systems I: Regular Papers, 67(2), 658-671 (2019).
- [2] Salamat, S., Imani, M., Gupta, S., & Rosing, T. Rnsnet: In-memory neural network acceleration using residue number system. In 2018 IEEE International Conference on Rebooting Computing (ICRC) (pp. 1-12) (2018).
- [3] Peng, J. et al. A Deep Neural Network Accelerator using Residue Arithmetic in a Hybrid Optoelectronic System. ACM Journal On Emerging Technologies In Computing Systems (JETC). 18, 1-26 (2022).
- [4] Bandyopadhyay, S., Sludds, A., Krastanov, S., Hamerly, R., Harris, N., Bunandar, D., Streshinsky, M., Hochberg, M. & Englund, D. A Photonic Deep Neural Network Processor on a Single Chip with Optically Accelerated Training. Conference on Lasers and Electro-Optics (CLEO) (pp. 1-2) (2023).
- [5] Pai, S. et al. Experimentally realized in situ backpropagation for deep learning in photonic neural networks. Science, 380(6643), 398-404 (2023).
- [6] Feng, C. et al. Integrated multi-operand optical neurons for scalable and hardware-efficient deep learning. Nanophotonics (2024).
- [7] Sun, J., Kumar, R., Sakib, M., Driscoll, J.B., Jayatileka, H. and Rong, H. A 128 Gb/s PAM4 silicon microring modulator with integrated thermo-optic resonance tuning. Journal of Lightwave Technology, 37(1), pp. 110-115 (2018).
- [8] Fujikata, J., Takahashi, S., Takahashi, M., Noguchi, M., Nakamura, T. and Arakawa, Y. High-performance MOS-capacitor-type Si optical modulator and surface-illumination-type Ge photodetector for optical interconnection. Japanese Journal of Applied Physics, 55(4S), p.04EC01 (2016).
- [9] Patel, D., Veerasubramanian, V., Ghosh, S., Samani, A., Zhong, Q. and Plant, D.V. High-speed compact silicon photonic Michelson interferometric modulator. Optics express, 22(22), pp. 26788-26802 (2014).
- [10] Han, J.H., Boeuf, F., Fujikata, J., Takahashi, S., Takagi, S. and Takenaka, M. Efficient low-loss InGaAsP/Si hybrid MOS optical modulator. Nature Photonics, 11(8), pp.486-490 (2017).
- [11] Kieninger, C. et al. Ultra-high electro-optic activity demonstrated in a silicon-organic hybrid modulator. Optica, 5(6), pp. 739-748 (2018).

- [12] He, M. et al. High-performance hybrid silicon and lithium niobate Mach–Zehnder modulators for 100 Gbit s⁻¹ and beyond. *Nature Photonics*, 13(5), pp.359-364 (2019).
- [13] Alexander, K., George, J.P., Kuyken, B., Beeckman, J. and Van Thourhout, D. Broadband electro-optic modulation using low-loss PZT-on-silicon nitride integrated waveguides. In *CLEO: Applications and Technology* (pp. JTh5C-7). Optica Publishing Group (2017).
- [14] Liu, J., Beals, M., Pomerene, A., Bernardis, S., Sun, R., Cheng, J., Kimerling, L.C. and Michel, J. Waveguide-integrated, ultralow-energy GeSi electro-absorption modulators. *Nature Photonics*, 2(7), pp.433-437 (2008).
- [15] Sorianello, V. et al. Graphene–silicon phase modulators with gigahertz bandwidth. *Nature Photonics*, 12(1), pp.40-44 (2018).
- [16] Eltes, F. et al. A BaTiO₃-based electro-optic Pockels modulator monolithically integrated on an advanced silicon photonics platform. *Journal of Lightwave Technology*, 37(5), pp.1456-1462 (2019).
- [17] Jenkins, W. K. Complex residue number arithmetic for high-speed signal processing. *Electronics Letters*, 16(17), 660-661 (1980).
- [18] Yen, S. M., Kim, S., Lim, S., & Moon, S. J. RSA speedup with Chinese remainder theorem immune against hardware fault cryptanalysis. *IEEE Transactions on computers*, 52(4), 461-472 (2003).
- [19] Murmann, B. Mixed-signal computing for deep neural network inference. *IEEE Transactions On Very Large Scale Integration (VLSI) Systems*. 29, 3-13 (2020).
- [20] Murmann, B. ADC performance survey 1997-2023. <http://web.stanford.edu/~murmman/adcsurvey.html>. (2011).
- [21] Verhelst, M., & Murmann, B. Area scaling analysis of CMOS ADCs. *Electronics letters*, 48(6), 1 (2012).
- [22] Hiasat, A. A residue-to-binary converter with an adjustable structure for an extended RNS three-moduli set. *Journal of Circuits, Systems and Computers*, 28(08), 1950126 (2019).
- [23] Rahin, A.B., Kadivar, A. and Rahin, V.B. Extremely High Frequency and Low Power Ring Oscillators Using DG-CNTFET Transistors. *IEEE 6th Conference on Technology In Electrical and Computer Engineering (ETECH)* (2021).

REVIEWERS' COMMENTS

Reviewer #2 (Remarks to the Author):

The authors have addressed my comments and the manuscript has been significantly improved. I support its publication in Nature Communications.

Reviewer #3 (Remarks to the Author):

The authors addressed the reviewer's comments sufficiently.